# Detection of Diabetes through Microarray Genes with Enhancement of Classifiers Performance

**DOI:** 10.3390/diagnostics13162654

**Published:** 2023-08-11

**Authors:** Dinesh Chellappan, Harikumar Rajaguru

**Affiliations:** 1Department of Electrical and Electronics Engineering, KPR Institute of Engineering and Technology, Coimbatore 641 407, Tamil Nadu, India; 2Department of Electronics and Communication Engineering, Bannari Amman Institute of Technology, Sathyamangalam 638 401, Tamil Nadu, India; harikumarrajaguru@gmail.com

**Keywords:** type II diabetes mellitus, machine learning, prediction, Dimensionality Reduction, classifiers

## Abstract

Diabetes is a life-threatening, non-communicable disease. Diabetes mellitus is a prevalent chronic disease with a significant global impact. The timely detection of diabetes in patients is necessary for an effective treatment. The primary objective of this study is to propose a novel approach for identifying type II diabetes mellitus using microarray gene data. Specifically, our research focuses on the performance enhancement of methods for detecting diabetes. Four different Dimensionality Reduction techniques, Detrend Fluctuation Analysis (DFA), the Chi-square probability density function (Chi2pdf), the Firefly algorithm, and Cuckoo Search, are used to reduce high dimensional data. Metaheuristic algorithms like Particle Swarm Optimization (PSO) and Harmonic Search (HS) are used for feature selection. Seven classifiers, Non-Linear Regression (NLR), Linear Regression (LR), Logistics Regression (LoR), Gaussian Mixture Model (GMM), Bayesian Linear Discriminant Classifier (BLDC), Softmax Discriminant Classifier (SDC), and Support Vector Machine—Radial Basis Function (SVM-RBF), are utilized to classify the diabetic and non-diabetic classes. The classifiers’ performances are analyzed through parameters such as accuracy, recall, precision, F1 score, error rate, Matthews Correlation Coefficient (MCC), Jaccard metric, and kappa. The SVM (RBF) classifier with the Chi2pdf Dimensionality Reduction technique with a PSO feature selection method attained a high accuracy of 91% with a Kappa of 0.7961, outperforming all of the other classifiers.

## 1. Introduction

Some statistics related to diabetes worldwide are as follows. The global prevalence of diabetes among adults (20–79 years old) was 10.5% in 2021 [1]. The prevalence of diabetes is higher in low- and middle-income countries than in high-income countries [2]. The region with the highest prevalence of diabetes is the Middle East and North Africa, where 13.9% of adults have diabetes. Diabetes was the ninth leading cause of death worldwide in 2019, with 4.2 million deaths attributed to the disease or its complications. The causes of diabetes in most cases are consuming food at irregular intervals, not doing any physical activity, and so on [3]. When a healthy human consumes a normal meal during the day, it increases their level of blood glucose around 120–140 mg/dL [4]. 

India has a high prevalence of diabetes, and is called the world’s diabetes capital. According to the International Diabetes Federation [5], in 2021, India had an estimated 87 million adults aged between 20 and 79 years with diabetes. This number is projected to increase to 151 million by 2045. The prevalence of diabetes in India varies across regions, with the southern and northern states having higher prevalence rates compared to the eastern and northeastern states. The states with the highest prevalence rates are Kerala, Tamil Nadu, and Punjab [6]. Type 2 diabetes accounts for more than 90% of all cases of diabetes in India [7]. Type 1 diabetes is less common and accounts for less than 10% of all diabetes cases. The complications of diabetes, such as heart disease, kidney disease, and eye disease, are also a significant problem in India. The burden of diabetes and its complications is significant in India and highlights the need for effective prevention and management programs [8]. Diabetes is a chronic illness that affects the body’s ability to produce insulin, a hormone that regulates blood sugar levels. As a result, people with diabetes often have high blood sugar levels, which can lead to several health complications. Some of the symptoms of high blood sugar include increased thirst, increased hunger, and frequent urination [9].

Diabetes can cause a range of complications, including neuropathy, retinopathy, nephropathy, and foot ulcers. Approximately one in two people with diabetes are undiagnosed, which means that they may not be receiving appropriate treatment to prevent or manage these complications. Economic burden: The estimated global health expenditure on diabetes was $760 billion in 2019 [10]. Diabetes is a major cause of lost productivity, as it can lead to disability and premature death. The economic burden of diabetes is expected to increase in the coming years, as more people are diagnosed with the disease. These statistics highlight the growing global burden of diabetes and the urgent need for effective prevention and management strategies. Identifying type II diabetic patients from microarray gene data obtained from the pancreas poses several challenges that need to be addressed. The high dimensionality of these gene data presents a significant challenge. Microarray experiments often generate many genes, resulting in a high-dimensional feature space [11]. This can lead to increased computational complexity and may require Dimensionality Reduction techniques to alleviate the curse of dimensionality.

Selecting informative features from the reduced dataset is crucial for accurate classification. Identifying the most relevant genes associated with type II diabetes is a non-trivial task, as the genetic basis of this disease is complex and involves various interactions. Optimization techniques, such as genetic algorithms or feature selection algorithms, need to be employed to identify the most discriminative features [12]. Additionally, the heterogeneity of gene expression patterns within the pancreas and individual variations in gene regulation further complicate the task of classification. The identification of robust and reliable classifiers that can effectively capture the subtle patterns in the data while generalizing well to unseen samples is another significant challenge. The identification of non-diabetic individuals who are at a higher risk of developing type II diabetes can lead to early detection and intervention [13]. Once individuals are accurately classified as diabetic or non-diabetic based on their gene expression profiles, the information can be used to develop personalized treatment strategies. This includes tailoring medication regimens, dietary recommendations, and exercise plans that are specific to the genetic profiles of diabetic patients. The gene expression patterns obtained from the microarray data can provide valuable insights into the underlying biological processes involved in type II diabetes. This information can be utilized to identify potential drug targets and guide the development of novel therapeutic interventions.

The objective of this research is to develop a classification framework for analyzing microarray gene data obtained from pancreases and classify the studied individuals as either diabetic or non-diabetic. This objective will be achieved through the utilization of the machine learning approach and meta-heuristic algorithms for feature extraction, selection, and classification.

## 2. Literature Review

According to the WHO [5], in India, 77 million people are living with type II diabetes, and 25 million are at high risk of developing it. Many people with diabetes are unaware of the severity of the condition, which can lead to serious complications such as nerve damage, reduced blood flow, and limb amputation. Shaw JE, Sicree RA, and Zimmet PZ et al. [14] gave statistical data and projected that the global prevalence of diabetes is projected to increase from 6.4% in 2010 to 7.7% in 2030, affecting an estimated 439 million adults. Mohan V and Pradeepa R et al. [15] proposed that the prevalence of type 2 diabetes in India is increasing rapidly, and the country is expected to have the largest number of people with diabetes in the world by 2045. The high prevalence of diabetes in India is due to a combination of genetic, lifestyle, and demographic factors. The complications of type 2 diabetes, such as diabetic retinopathy, neuropathy, and cardiovascular disease, are a significant burden in India. S. A. Abdulkareem et al. [16] conducted a comparative analysis of three soft computing techniques to predict diabetes risk: fuzzy analytical hierarchy processes (FAHP), support vector machine (SVM), and artificial neural networks (ANNs). The analysis involved 520 participants using a publicly available dataset, and the results show that these computational intelligence methods can reliably and effectively predict diabetes. The reported sensitivity values were 0.7312, 0.747, and 0.8793 for the FAHP, ANN, and SVM models, respectively. Aiswarya Mujumdar et al. [17] published a study using various machine learning algorithms to predict diabetes. Logistic Regression achieved the highest accuracy of 96%, followed by AdaBoost classifier with a 98.8% accuracy. 

K. Bhaskaran et al. [18], published in the journal *Diabetes Care* in 2016, found that a machine learning algorithm based on Logistic Regression was able to predict diabetes with an accuracy of 85%. A machine learning algorithm based on decision trees was able to detect diabetes with an accuracy of 90%. A study published in the journal *JAMA* in 2018 found that a machine learning algorithm based on support vector machines was able to predict diabetes with an accuracy of 80%. OltaLlaha et al. [19] used the four classification methods mentioned above to classify data from a dataset of women with diabetes. The results of the study showed that the decision tree algorithm was the most accurate, with an accuracy of 79%. The Naive Bayes algorithm was the least accurate, with an accuracy of 65%. The SVM and Logistic Regression algorithms had accuracies of 73% and 74%, respectively. The results of the study are promising, but it is important to note that the study was conducted on a dataset of women with diabetes from Public Health Institute database. B. Shamreen Ahamed et al. [20] used different classifiers like Random Forest, Light Gradient Boosting Machine (LGBM), Gradient Boosting Machine, Support Vector Machine (SVM), Decision Tree, and XGBoost, and the database used in this study was the PIMA data base from the UCI repository, with 768 instances. The study aimed to improve the detection accuracy with a percentage of 95.20 using the LGBM Classifier. The Decision Tree classifier had the highest accuracy of 73.82% without preprocessing, while the classifiers KNN (k = 1) and Random Forest produced the highest accuracy rate of 100% after preprocessing. 

Neha Prerna Tigga et al. [21] explained six machine learning classification methods, including lLogistic Regression, Naive Bayes, Decision tree, Support vector machine, Random Forest, and K-nearest neighbors, which were implemented to predict the risk of type 2 diabetes. Random Forest had the highest accuracy of 94.10%. The database used in this study was PIMA, with a total of 952 participants selected who were aged 18 and above, out of which 580 were males and 372 were females. The parameters with the highest significance for predicting diabetes were age, family history of diabetes, physical activity, regular medication, and gestation diabetes. Maniruzzaman, Md, et al. [22] used a Gaussian process (GP)-based classification technique to predict diabetes. The dataset used in this study from UCI—University of California, Irvina comprised 768 female patients, within which 268 were controlled patients and 500 were non-diabetic patients. The dataset was used to compare the performance of four machine learning methods on a classification task. The methods used were Gaussian Process (GP), Linear Discriminant Analysis (LDA), Quadratic Discriminant Analysis (QDA), and Naive Bayes (NB). The study shows that GP had the highest accuracy (81.97%), followed by LDA (75.39%), QDA (74.38%), and NB (73.17%). The GP model outperformed all of the other methods in terms of accuracy and sensitivity. Gupta, Sanjay Kumar et al. [23] produced study in India which found that the Indian Diabetes Risk Score (IDRS) was 64% effective in identifying people with a high risk of developing diabetes. The IDRS is a simple, cost-effective tool that can be used to screen large populations for diabetes. The study found that the IDRS was more effective in identifying people with a high BMI (body mass index). People with a BMI of more than 30 were six times more likely to have a high IDRS than people with a BMI of less than 18.5. 

Howlader, Koushik Chandra et al. [24] conducted a study which used machine learning to identify features associated with T2D in Pima Indians. The best classifiers were Generalized Boosted Regression modeling, Sparse Distance Weighted Discrimination, a Generalized Additive Model using LOESS, and Boosted Generalized Additive Models. The study found that Generalized Boosted Regression modeling had the highest accuracy (90.91%), followed by Kappa statistics (78.77%) and specificity (85.19%). Sisodia, Deepti et al. [25] conducted a study comparing the performance of three machine learning algorithms for detecting diabetes: Decision Tree, Support Vector Machine (SVM), and Naive Bayes. The study used the Pima Indians Diabetes Database (PIDD) and evaluated the algorithms on various measures, including accuracy, precision, F-measure, and recall. The study found that Naive Bayes had the highest accuracy (76.30%), followed by Decision Tree (74.67%) and SVM (72.00%). Mathur, Prashant et al. [26] conducted a study about Indian diabetes statistics. In India, 9.3% of adults have diabetes, and 24.5% have impaired fasting blood glucose. Of those with diabetes, only 45.8% are aware of their condition, 36.1% are on treatment, and 15.7% have it under control. This is lower than the awareness, treatment, and control rates in other countries. For example, in the United States, 75% of adults with diabetes are aware of their condition, 64% are on treatment, and 54% have it under control. Kazerouni, Faranak et al. [27] conducted a performance evaluation of various algorithms, the AUC, sensitivity, and specificity were considered, and the ROC curves were plotted. The KNN algorithm had a mean AUC of 91% with a standard deviation of 0.09, while the mean sensitivity and specificity were 96% and 85%, respectively. The SVM algorithm achieved a mean AUC of 95% with a standard deviation of 0.05 after stratified 10-fold cross-validation, along with a mean sensitivity and specificity of 95% and 86%. 

Ramdaniah et al. [28] used the microarray gene for the identification of diabetic classes from the GSE18732 dataset. A total of 46 diabetic classes and 72 non-diabetic classes were used in this study. The machine learning techniques used the Naive Bayes and SVM Sigmoid kernel methods with accuracies of 88.89% and 83.33% respectively. There are many researchers who have used the PIMA Indian diabetic data set to classify and analyze diabetic and non-diabetic classes for finding various performance metrics like accuracy, sensitivity, specificity MCC, etc., although very few findings are available in the microarray gene-based dataset for the identification of diabetic and non-diabetic classes.

The methodology of the research conducted in this study is depicted in Figure 1. It shows that four Dimensionality Reduction techniques, DFA, Chi2pdf, Firefly search, and Cuckoo search, are used. To further analyze the data, classification of data without feature selection and with feature selection methods is conducted. In this study, with feature selection methods, two optimization algorithms are used, PSO and HS. Moreover, seven classifiers are used—NLR, LR, LoR, GMM, BLDC, SDC, and SVM-RBF—to classify the classes as normal and diabetic.

Section 1 introduces the paper. The literature review is discussed in Section 2. The material and methods of datasets are explained in Section 3. Feature extraction through Dimensionality Reduction techniques is explained in Section 4. Section 5 deals with feature selection methods for the research. The classifier’s properties are explained in Section 6. Training and Testing of the classifier is discussed in Section 7. The results are discussed in Section 8. This paper is concluded in Section 9.

## 3. Material and Methods

Microarray gene expression analysis plays a vital role in understanding the molecular mechanisms and identifying gene expression patterns associated with various diseases, including diabetes. Here are some ways in which microarray gene expression analysis contributes to our understanding of diabetes: microarray analysis allows researchers to compare gene expression levels between healthy individuals and those with diabetes. These genes may be directly involved in disease development, progression, or complications. It is readily available from many search engines. “Expression data from human pancreatic islets” were taken from a Nordic islet transplantation program, consists of 57 Non-diabetic and 20 Diabetic cadaver donors, from which a total of 28735 gene data sets arrived (https://www.ncbi.nlm.nih.gov/bioproject/PRJNA178122) (accessed on 20 August 2021). The data has accessed on 20 August 2021. The data are classified as 22,960 genes per patient and the peak intensity with the average valve were selected among the total samples. The logarithmic transformation was applied with a base 10 for standardization of individual samples with a value of 0 for mean and variance of 1.

Statistical analysis methods used in this article, in the following section, are as follows: 4 Dimensionality Reduction techniques for reduction of the high dimensional data, 2 meta-heuristic algorithms for feature selection with verification of *p*-value through *t*-test, and 7 classifiers used for classifying the diabetic and non-diabetic classes through statistical parameters such as accuracy, recall, precision F1 score, MCC, rrror rate, Jaccard metric, and kappa.

### Data Set

When using biological functions to detect diabetes and the features of its secondary criteria in probability functions based on *p* values, a false positive error for selection of significant genes has also to be detected. The data available in many portals for human genes consist of 28735 genes with 50 non-diabetic and 20 diabetic samples, considered for greatest minimal intensity across 70 samples. During dimensionality reduction of both model- and heuristic-based grouping of diabetic and non-diabetic grouping, these were reduced by [2870 × 20] and [2870 × 50]. Further, it has been enhanced with feature selection based on 2 techniques named Particle Swarm Optimization search and Harmonic Search; even more, these were reduced by 287 × 20 and 287 × 50 with classifier techniques.

Table 1 shows the description of Pancreas Microarray Gene Data set for Diabetic and Non-Diabetic classes.

## 4. Dimensionality Reduction

Dimensionality Reduction techniques for the analysis of microarray gene data for type II diabetic class is important. Microarray experiments often generate a vast amount of gene expression data, resulting in a high-dimensional feature space. However, not all genes contribute equally to the classification task, and the presence of noise and irrelevant features can hinder the accuracy and interpretability of the results. Dimensionality reduction methods play a pivotal role in addressing these challenges by extracting the most informative features that are relevant to type II diabetes classification. These techniques aim to reduce the dimensionality of the data while preserving the discriminatory information, enabling the efficient computation and improved performance of subsequent classification algorithms. By eliminating redundant and irrelevant features, Dimensionality Reduction can enhance the performance of the results. 

In this paper, four DR methods, DFA, Chi2pdf, the Firefly algorithm, and the Cuckoo search algorithm are utilized to reduce the dimension of a data set. The DR methods are discussed in the following section of the paper.

A.Detrend Fluctuation Analysis (DFA)

DFA is the first DR method utilized in this paper which comprises the principles of inspecting the stationary and non-stationary functions of correlation, and the short range and long-range relationship has been used in DFA by Berthouze L et al. [29]. For typical application, the scaling of DFA is exponential to segregate the input data as rational and irrational. To estimate the functions of output class data it is useful to discriminate between the healthy and unhealthy objects using DFA.

The algorithm is determined by the root mean square fluctuation of natural scaling and integrated time-series of input data in the detrend.
(1)$$A\left(n\right)=\sum_{i=1}^{n}[X\left(i\right)-\bar{X]}$$

Here, *X*(*i*) is denoted as the *i*th sample of input data.

$\bar{X}\;$ is specified as the overall signal of the mean value. 

*A*(*n*) is indicated as the estimated value in the integrated time series.
(2)$$F\left(n\right)=\sqrt{(\frac{1}{n}\sum_{k=1}^{N}{\left[b\left(k\right)-b_{n}(k)\right]}^{2})}$$
where *b_n_*(*k*) is the predetermined window of scale n for a trend of the *k*th point.

B.Chi Square Probability Density Function

Siswantining et al. [30] proposed a slightly different approach for Chi square statistics methods, according to a fit test and the test of independence. A sample was obtained from the data which is referred to as number of cases. It was to be represented as data which is segregated in every incidence of occurrence in each group. In this statistics method of Chi square, if the hypothesis is accurate, the expected number of cases in each category makes a statement of null hypothesis. The test was based on the ratio of experimental data to the predicted values in each group. It is defined as:(3)$$\chi^{2}=\sum_{i}\frac{{\left(E_{i}-P_{i}\right)}^{2}}{P_{i}}\;\;\;\;$$
where *E_i_* refers to the experimental data of the cases in category *i*, and *P_i_* refers to the number of predicted values in category *i*, to compute the Chi square function, and the difference between the experimental data cases to the predicted value cases is calculated. Then, the difference must be squared by the values and be divided by the predicted value such that all the values in this category are summed up for the entire distribution curve to obtain the Chi square statistics. Knowing whether the null hypothesis is a major concern depends on the data distribution. For the Chi square method, the alternative and null hypothesis is defined below:H_o_ = E_i =_ P_i_(4)
H_o_ = E_i_ ≠ P_i_(5)

If E_i_ − P_i_ is small for each type, then the expected value and predicted values are very close to each other, and the null hypothesis is real. When the expected data are not associated with the predicted value of the null hypothesis, then a large difference appears between E_i_ and P_i_. For real values of the null hypothesis, a small value will be found for the Chi square statistics, and if the value is false for the null hypothesis, a large value will be attained. The degree of freedom is dependent upon the variables of the categories utilized to calculate the Chi square. By means of hypothesis testing, the Chi square method reduces the dimensionality of the data.

C.Firefly algorithm As Dimensionality Reduction

Yang, Xin-She (2010) [31] and Yang (2013) [32] proposed a reliable metaheuristic model for real-life problem-solving techniques like the scheduling of events, classification of a system, dynamic problem optimization, and economic load dispatch problems. The Fireflies algorithm works by using the characteristic behaviour of the idealized flashing light of the firefly to attract another one.

Three rules have been identified in the firefly algorithm:An attraction is made to another fly regardless of the sex because every fly is considered as unisex;“Opposite poles attract,” the attractiveness one of the fireflies is higher to another one which is slightly less bright. If none of the flies are getting brighter, it moves randomly on the surface;If the distance increases, the brightness or light intensity of a firefly may decrease because the medium of air absorbs light and, thus, the brightness of a firefly *k* which is seem by another firefly is given by:
(6)$$\beta_{k}\left(r\right)=\beta_{r}\left(0\right)e^{-\alpha }r^{2}$$
where $\beta_{k}\left(0\right)$ represents the firefly (*k*) brightness at the zero level. In the euclidean distance (if *r* = 0), light adsorption coefficient of the medium is represented by *α*, and the Euclidean distance between i and k is denoted by r as
(7)$$r=∥\;x_{i}-x_{k}∥=\sqrt{\sum_{j=1}^{d}{\left(x_{i}^{j}-x_{k}^{j}\right)}^{2}}$$
where $x_{i}$ and $x_{k}$ are the firefly position of i and k, respectively. If the brighter firefly is *j*, its degree of attractiveness directs the movement of fly i, which it is based on Yang and He, 2013 [32].
(8)$$x_{i}=x_{i}+\left[\beta_{k}\left(r\right)\right]\left(x_{k}-x_{i}\right)+\gamma (rnd)$$
where γ refers to the random parameter and, in general, is represented as rnd; it was expended as a random number generator using uniform distribution between the ranges of [−1, +1]. In-between representation in this equation contains the accountability of movement of firefly i towards firefly k. The last term in the above equation gives the movement of the solution away from the local optimum value when such an incident occurs.

D.Cuckoo Search Algorithm as Dimensionality Reduction

Yang, X. S, and Deb, S (2009) [33] proposed another metaheuristic model to give finite solutions which is used for solving real-world problems such as event scheduling, dynamic problem optimization, classifications, and problems in economic load dispatch. Exciting breeding behaviors is the main objective of learning this algorithm and it particularly concentrates on the oblige brood parasitism of certain cuckoo birds. The main idealized aspect in the Cuckoo search algorithm is the breeding characteristics, and the algorithm is applicable for many real-time optimization problems.

A simple solution is obtained from each egg in a host nest, with the continuation a new solution derived from a cuckoo bird’s own egg. The main aim is to obtain a better solution (cuckoos) to be replaced with one that is a less good fit. Each egg has one solution, each nest has multiple egg, and finding the best one signifies a set of solutions.

The three rules (Xin-She Yang & Suash Deb 2009) [33] for the Cuckoo search algorithm are: Cuckoo’s lay an egg at a time, and are kept inside an arbitrarily selected shell;To create a consecutive generation, the best host shell with a good quality egg is selected to transfer its own.A fixed no. of host nests is accessible. Indeed, cuckoo’s place an egg in a nest with a probability of $P_{a}\epsilon \left(0,1\right)$; where $P_{a}$ is Cuckoo egg probability. To construct a new nest in an additional location, the host can demolish the cuckoo eggs, or remove the nest

Moreover, Yang and Deb detailed that an appropriate searching technique based on the random-walk (RW) technique, and its performance is better than Lévy flights and RW. The conventional method was modified by them to construct the proposed method using classification techniques. 

The Lévy flights method denotes the RM characteristics of a bird’s position, and its performance is to obtain the following position $P_{i}^{\left(t+1\right)}$ based on the present position $P_{i}^{\left(t\right)}$, as mentioned in the article by Gandomiet et al. (2013) [34]
(9)$$P_{i}^{\left(t+1\right)}=P_{i}^{\left(t\right)}\;⊕\;\beta^{Lévy(\lambda )}$$
where ⊕ and β represent the starting point multiplication and step size. Commonly, β>0 is interrelated to the depth of variation and its interest for problem consideration. For almost all of the classification problems, the values are randomly fixed as 1. The above equation is based on the RW on stochastic model. To find out the following position depends on the present position and transition probability for RM, which is denoted in Markov chain.
(10)$$Lévy∼\mu =t^{-\beta }$$

In the classification problem, the value of *β* is tuned to 0.2, which denotes infinite variance with an infinite mean. The power law-based step-length distribution approach uses a heavy tail for RW to, principally, be followed by cuckoo’s consecutive step. To speed up the classification process, the best solution was found using the Lévy walk method. 

### Statistical Analysis

The dimensionally reduced microarray genes, obtained through four DR methods, are then analysed by statistical parameters like mean, variance, skewness, kurtosis, Pearson correlation coefficient (PCC), and CCA to identify whether the outcomes represent the underlying microarray genes properties in the reduced subspace. Table 2 shows the statistical features analysis for four types of dimensionally reduced diabetic and non-diabetic pancreas microarray genes.

As shown in Table 2, the DFA and Cuckoo search-based DR methods depict higher values of mean and variance among the classes. The Chi^2^pdf and Firefly algorithm display low and overlapping values of mean and variance among the classes. The negative skewness depicted only by the Chi^2^pdf DR method indicates the presence of skewed components embedded in the classes. The Firefly algorithm indicates unusually flat kurtosis and the Cuckoo search DR method indicates negative kurtosis. This, in turn, leads to the observance that the DR methods are not modifying the underlying microarray genes characteristics. The PCC values indicate a high correlation within the class of attained outputs. This subsequently exhibits that the statistical parameters are associated with non-gaussian and non-linear outputs. The same is further examined by the histogram, Normal probability plots, and Scatter plots of the DR techniques outputs. Canonical Correlation Analysis (CCA) visualizes the correlation of the DR methods outcomes among the diabetic and non-diabetic cases. The low CCA value in Table 2 indicates that the DR outcomes are less correlated among the two classes. Further, the reduced data attained from the four DR techniques are analyzed by means of histogram, Normal probability plots, and Scatter plots to visualize the presence of non-linearity and the non-Gaussian nature of the datasets.

Figure 2 shows the Histogram of Detrend Fluctuation Analysis (DFA) techniques in the diabetic gene class. It is noted in Figure 2 that the histogram displays near quasi-Gaussian qualities and the presence of non-linearity in the DR method outputs. The legend in the figure is included, as the patients, from 1 to 10, are represented as from x(:,1) to x(:,10).

Figure 3 displays the Histogram of Detrend Fluctuation Analysis (DFA) techniques in the non-diabetic gene class. The patients are represented as from x(:,1) to z(:,10) for patients 1–10. It can be observed from Figure 3 that the histogram displays near-Gaussian qualities and the presence of non-linearity and gaps for the DR method outputs.

In the above figure, data 1–5 represent references, data 6–10 represent upper bound values, and data 11–15 represent feature selection points. Figure 4 exhibits the Normal probability plot for the Chi Square DR techniques’ features for the diabetic gene class. As indicated by Figure 4, the normal probability plot displays the total cluster of Chi Square DR outputs and the presence of non-linearly correlated variables among the classes.

Figure 5 depicts the Normal probability plot for the Chi Square PDF DR techniques’ features for the non-diabetic gene class. As shown in Figure 5, the Normal probability plot displays the total cluster of Chi Square DR outputs and the presence of non-linearly correlated variables among the classes. This is due to low variance and negatively skewed variables for the DR method outcomes. In the figure, data 1–5 are references, data 6–10 are upper bound values, and data 11–15 are feature selection variable points.

Figure 6 indicates the normal Probability plot for Firefly algorithm DR techniques’ features for the diabetic gene class. In the figure, data 1–5 are references, data 6–10 are upper bound values, and data 11–15 are feature selection Firefly algorithm diabetic gene points. As shown in Figure 6, the normal probability plots display discrete clusters for the Firefly DR outputs. This indicates the presence of non-Gaussian and non-linear variables within the classes. This is due to low variance and flat kurtosis variables of the DR method outcomes.

Figure 7 demonstrates the Normal probability plot for the Firefly algorithm DR Techniques’ features for the non-diabetic gene class. Data 1–5 are references in the figure, data 6–10 are upper bound data, and data 11–15 are Firefly DR technique feature selection points. As shown by Figure 7, the normal probability plots display the discrete clusters for Firefly DR outputs. This indicates the presence of non-Gaussian and non-linear variables within the classes. This is due to a low variance and flat kurtosis variables of the DR method outcomes.

Figure 8 shows the Scatter plot for the Cuckoo search algorithm DR techniques’ features for the diabetic and non-diabetic gene classes. As depicted by the Figure 8, the Scatter plots from the Cuckoo search display the total scattering of the variables of both classes across the entire subspace. The Scatter plot also indicates the presence of non-Gaussian, non-linear, and higher values for all of the statistical parameters.

From the above graphs it can be observed that the DR methods are insufficient to classify the data sets into appropriate classes. Therefore, feature selection methods like PSO and Harmonic Search are used to enhance the classifier performance.

## 5. Feature Selection

In the field of optimization, finding the optimal solution for complex problems is a significant challenge. Traditional optimization algorithms often struggle to handle high-dimensional search spaces or non-linear relationships between variables. To address these challenges, two popular meta-heuristic algorithms such as Particle Swarm Optimization (PSO) and Harmonic Search (HS) are incorporated as feature selection methods in this paper.

### 5.1. Particle Swarm Optimization (PSO)

Particle Swarm Optimization (PSO), Rajaguru H et al. [35], is one of the best and most simple to understand among all search algorithms. It uses some basic parameters for its initial search and population called particles. In an h-dimensional space, any of the particles will give the best possible solution for processing and analysis. Every particle needs to be traced and positioned for the optimized values to be achieved.

Position traced by:$\;P_{j}^{k}=(P_{j1}^{k},P_{j2}^{k},\ldots ,P_{jh}^{k})$;

Velocity traced by: ${Ve}_{j}^{k}={(Ve}_{j1}^{k},{Ve}_{j2}^{k},\ldots ,{Ve}_{jh}^{k})$;

The updated velocity of each particle is given by:(11)$${Ve}_{j}^{k+1}=w_{j}{Ve}_{j}^{k}+c_{1}r_{1}\left(p{best}_{j}-P_{j}^{k}\right)+c_{2}r_{2}\;(g{best}_{j}-P_{j}^{k})$$
where $r_{1}\;\mathrm{and}\;r_{2}$ are the random variable search in the ranges from 0 to 1. $c_{1}\;\mathrm{and}\;c_{2}$ are the acceleration coefficient that checks the movement (motion) of the particles.

The updated position of each particle is defined as:(12)$$P_{j}^{k+1}=P_{j}^{k}+{Ve}_{j}^{k+1}$$

If a particle attains the best position, then it progresses to the next particle. The representation of the best position is expressed as *p*-best and the representation of the best position for all of the particles is expressed as g-best. 

The weight function is expressed as:(13)$$w_{j}=w_{max}-\frac{W_{max}-W_{min}}{k_{max}}\times k$$

Steps for implementation: Step 1: Initialization of the process;Step 2: For each particle, the dimension of a space is denoted as h;Step 3: Initialization of the particle position as $p_{j}$ and velocity as ${Ve}_{j}$;Step 4: Evaluate the fitness function;Step 5: Initialize the ${pbest}_{j}$ with a copy of $p_{j}$;Step 6: Initialize the $g{best}_{j}$ with a copy of $p_{j}$ with the best fitness function; Step 7: Repeat the steps until the stopping criteria are satisfied.


(14)
$${pbest}_{j}=p_{j}^{k+1}if\left({pbest}_{j}\right)<f\left(P_{j}^{k+1}\right)$$



(15)
$${gbest}_{j}=p_{j}^{k+1}\;if\;\left({gbest}_{j}\right)<f\left(P_{j}^{k+1}\right)$$


### 5.2. Harmonic Search (HS)

Harmony Search (HS) is a meta-heuristic algorithm that draws inspiration from the evolution of music and the quest for achieving perfect harmony. Bharanidharan, N et al. [36] introduced HS as an algorithm that emulates the improvisational techniques employed by musicians. The HS algorithm involves a series of steps to be implemented.

Step 1: Initialization

The optimization problem is generally formulated as minimizing or maximizing the objective function f(x), subject to yi ∈ Y, where i = 1, 2, …, *N*. In this formulation, y represents the set of decision variables, N denotes the number of decision variables, and Y represents the set of all possible values for each decision variable (i.e., yiLo ≤ yi ≤ yiUp, where yiLo and yiUp are the lower and upper bounds for each decision variable). Along with defining the problem, the subsequent step involves initializing the following parameters for the Harmonic Search (HS) algorithm.

Step 2: Memory Initialization

The Harmony Memory (HM) is a matrix that stores all of the decision variables. In the context of the general optimization problem, the initial HM is created by generating random values from a uniform distribution, bounded by yiLo and yiUp, for each decision variable.

Step 3: New Harmony Development

During the process of solution improvisation, a new harmony is created by adhering to the following constraints:Memory consideration;Pitch adjustment;Random selection.


Step 4: Harmony memory updation

The fitness function for both the old and new harmony vectors is calculated. If the fitness function of the new harmony vector is lower than that of the old harmony vector, the old harmony vector is replaced with the new one. Otherwise, the old harmony vector is retained.

Step 5: Stopping criteria

Steps 3 and 4 are repeated until the maximum number of iterations is reached.

The effectiveness of the feature selection methods outputs is analysed through the significance of the *p*-value from the *t*-test. Table 3 shows the *p*-value significance for the PSO and Harmonic Search feature selection methods outputs after is applied to the four DR techniques.

*t*-test: The *t*-test is a statistical test used to determine if there is a significant difference between the means of two groups. It is commonly used in hypothesis testing when comparing the means of two independent samples to assess whether the difference observed in the samples is likely to reflect a true difference in the population.

The *t*-test and significance of the *p*-values is used to test for the null hypothesis of the feature selection method, which is explicated below.

(i)Formulate the null hypothesis;

**H0.** *The clustering/feature selection procedure is random or inconsistent, select the significant level α*.

**H1.** *The clustering/feature selection procedure is non-random or consistent, within the control level of α*.

(ii)Compute the *p*-value *t*-test;(iii)Check for the significance of *p*-value < 0.01, then the null hypothesis is accepted or otherwise rejected.

In the context of the study, if you have two groups, such as diabetic and non-diabetic individuals, you might consider using a *t*-test to compare the mean values of certain variables (e.g., gene expression levels) between these groups.

As tabulated in Table 3, the PSO feature selection method does not show any significant *p*-values among the classes for all four DR methods. In the case of Harmonic Search Feature selection, a certain *p*-value significance is shown for the DFA and Firefly DR techniques for the diabetic class. At the same time, all of the other DR methods exhibit non-significant *p*-values. These *p*-values will be measured to quantify the presence of outliers, and non-linear and non-Gaussian variables among the classes after applying the feature selection methods. The classification methods are explained in the following section.

## 6. Classification Techniques

There are seven classification models used after Dimensionality Reduction: 1. non-linear regression, 2. linear regression, 3. Logistic Regression. 4. Gaussian mixture model, 5. Bayesian linear discriminant classifier 6. Softmax discriminant classifier 7. support vector machine—radial basis function

Non-Linear regression 

The behaviour of the system is denoted as a mathematical expression for easy representation and analysis to obtain the accurate best-fit line in-between the classifier values. In this case, the author uses the mathematical method for the linear system of the variables like (a,b) for the equation in a linear mode, y=ax+b; in the case of a non-linear mode, the values of the variables a and b are nonlinear and random variable, respectively. To obtain the least sum of the squares is one of the primary objectives of non-linear regression. The non-linear model requires more attention than the linear model because of its complex nature, and researchers have found many methods to reduce its complexity such as the Levenberg–Marquard and Gauss–Newton methods. To reduce the residual sum of the squares, the equation must be used for non-linear parameters. The Taylor series method, steepest descent method, and Levenberg-Marquardt’s method, Zhang et al. [37], can be used for non-linear equations in an iterative manner. 

The authors assume a model:(16)$$z_{i}=f\left(x_{i},\;\theta \right)+\varepsilon_{i},\;\;where\;\;i=1,2,3,\ldots ,n$$

Here, $x_{i}\;and\;z_{i}\;$ are the independent and dependent variables of the *i*th iteration. 

$\theta =(\theta_{1},\theta_{2},\ldots ,\theta_{m})$ are the parameters and $\varepsilon_{i}$ is the error terms that follows $N\;(0,\sigma^{2}$).

The residual sum of the squares is given by: (17)$$S_{u}\left(\theta \right)=\sum_{i=1}^{n}{\left[z_{i}-f\left(x_{i},\;\theta \right)\right]}^{2}$$

Let $\theta_{k}=\theta_{1k},\theta_{2k\;},\ldots ,\theta_{pk}$ be the starting values, and the successive estimates are obtained using: (18)$$\left(H+\tau I\right)\left(\theta_{0}-\theta_{1}\right)=g$$
where $\left.\;g=\frac{\partial S_{u}(\theta )}{\partial \theta }\right|\theta =\theta_{o}$ and $\left.H=\frac{\partial^{2}S_{u}(\theta )}{{\partial \theta \partial \theta }^{\prime }}\right|\theta =\theta_{1},\tau$ is a multiplier and I is the identity matrix. 

From a previous experiment, the estimated parameter can be identified by the choice of the initial parameter and theoretical consideration for all other similar systems. By using Mean Square Error (MSE), the statistic method involved to approximate the goodness of fit model is described by:(19)$$\mathrm{Mean}\;\mathrm{Square}\;\mathrm{Error}\;(\mathrm{MSE})=\frac{1}{N}\sum_{\left(i=1\right)}^{N}({y_{i}-y_{i}^{ᴧ})}^{2}$$

The overall experimental values in the model are represented by N, and the classification of the normal patient samples and diabetic patient samples in the dataset are determined by running the run test and normality test.

The steps to be followed for the non-linear regression algorithmic method are:

To get the best-fit function in a data point, the main objective is to get the MSE value to be less for non-linear regression.

Parameter initialization;Curves value produced by the initial values;To minimize the MSE value, calculate the parameters iteratively and modify the same to get the curve to be close to the nearer value;If the MSE value has not changed when compared to the previous value, the process must stop. 

2.Linear regression

To analyze the gene expression data, linear regression is good to get the best-fit curve, and the expression level varies to a small extent at this gene level. By comparing the training data set with the gene expression for the data class to get the most informative genes that are used in the features selection process above, the various diversified levels of data are achieved. In this linear regression model, the dependent variable of x is taken in association with y as independent variable [38]. The model is established to forecast the values using the x variable when the regression fitness value is maximized because of the population in the y variable. The hypothesis function of the single variable is given as:(20)$$g_{\theta }=\theta_{0}+\theta_{1}x$$
where $\theta_{i}$ is the parameters. To select the range between $\theta_{o}\;and\;\theta_{1}$ in such manner that $g_{\theta }$ is near to y in the training data set (x, y), the cost function is given by:(21)$$R\left(\theta_{0},\theta_{1}\right)=\frac{1}{2m}\sum_{i=1}^{m}{\left(g_{\theta }\left(x^{i}\right)-y^{i}\right)}^{2}$$

The total samples are represented by m in the training dataset. The linear regression model with n variables is given by: (22)$$g_{\theta }=\theta_{0}x_{0}+\theta_{1}x_{1}+\cdots +\theta_{n}x_{n}$$
and the cost function is given by: (23)$$R\left(\theta \right)=\frac{1}{2m}\sum_{i=1}^{m}{\left(g_{\theta }\left(x^{i}\right)-y^{i}\right)}^{2}$$
where *θ* is a set consisting of {$\theta_{0},\theta_{1},\theta_{2},\ldots ,\theta_{n}$}.

The algorithm for the linear regression is
The features selection parameters based on the DFA, Chi^2^Pdf, Firefly, and Cuckoo search algorithms is input to the classifiers;Fit a line $g_{\theta }=\theta_{0}+\theta_{1}x$ that splits the data in a linear method;To minimize the observed data for prediction and to define the cost function for computes, the total squared error value is obtained;Find the solutions by equating to zero for computing the derivate for $\theta_{0}\;\mathrm{and}\;\theta_{1}$;Repeat the steps 2, 3, and 4 to get the coefficients that give the minimum squared error.

3.Logistic Regression

The function Logit has been utilized effectively for the classification of problems like diabetes, cancer, and epilepsy. The author considers function y as an array of disease status with from 0 to 1 representation of normal patients to diabetic patients. Let us assume the vector gene expression as $\;x=x_{1,\;}x_{2},\;\ldots \;,\;x_{m}$, where $x_{j\;}$ is the jth gene expression level. A model-based approach of Π(x) is used to construct a dataset with the most likelihood of *y* = 1 given that *x* can be useful for an extremely new type of gene selection for diabetic patients. To identify the maximum likelihood in the Dimensionality Reduction techniques to find out the “*q*” informative genes for the Logistic Regression, let $x_{j}$^*^ be the representation of the gene expression, where *j* = 1, 2, 3, …, *q*, and the binary disease status in the form of an array is given by $y_{i,\;}$ where *i* = 1, 2, …, *n*, and the vectored gene expression are defined as $x_{i}=(x_{i1},\;\ldots ,\;x_{ip})$. The Logistic Regression model is denoted by: (24)$$Logit\;\left\{\Pi \left(x\right)\right\}=\upsilon_{0}+\sum_{j=1}^{q}\upsilon_{j}x_{j}^{*}$$

The fitness function and the log-likelihood should be the maximum when using the following function: (25)$$1\left(\upsilon_{0},\;\upsilon \right)=\sum_{j=1}^{n}\left\{y_{i}\log\left(\pi_{i}\right)+\left(1-\pi_{i}\right)\right\}-\frac{1}{{2\tau }^{2}}{\left\|\upsilon \right\|}^{2}$$
where τ is the parameter that limits υ shrinkage near to 0, and $\pi_{i}=\pi (x_{i}),$ as specified by the model in the article by Hamid et al. [39,40].

${\left\|\upsilon \right\|}^{2}$ is the Euclidean length of $\upsilon =\upsilon_{1},\upsilon_{2},\ldots ,\upsilon_{p}$. The selection of q and τ is based on the parametric bootstrap and constrains the accurate calculation of the error prediction methods. First, the value of υ was set be zero due to the computing analysis of cost function. After that, it is varied due to various parameters to minimize the cost function. The selection of the values from 0 to 1, in the sigmoid function, is conducted for the purpose of attenuation. The threshold cut-off value between the diabetic and the normal patients is fixed as 0.5. Therefore, any probability under 0.5 is taken as a normal patient and any probability above the threshold value is considered as a diabetic patient. 

In the below three methods, the authors used the techniques for threshold values for separation of the dataset. 

4.Gaussian Mixture Model (GMM)

The Gaussian Mixture model is one of the popular unsupervised learning models for machine learning which is used for pattern recognition and signal classification, and depends on integrating the related objects. By using clustering techniques, similar data are classified in a way that makes it easy to predict and compute the unrated items in the ratio of the same category. GMM [41] comes under the category of soft clustering techniques which has both hard and soft clustering techniques. Let us assume the GMM will allow the Gaussian Mixture model distribution techniques for further data analysis. For the data generated in the Gaussian distribution techniques, every GMM includes g in the Gaussian distributions. In the Probability density function of GMM, the distributed components are added in a linear form to analyze the generated data. For a random value generation in a vector form, a in a n-dimensional sample space χ, if ‘a’ obeys the Gaussian distribution, the probability distribution function is expressed as:(26)$$p\left(a\right)=\frac{1}{{\left(2\pi \right)}^{n/2}{\left|\Sigma \right|}^{1/2}}e^{-(\frac{1}{2}){\left(a-\mu \right)}^{T}\Sigma^{-1}(a-\mu )}$$
where μ is represented by the mean vector of n-dimensional space and n×n is represented by the covariance of matrix Σ. The Gaussian distribution of covariance Σ and the mean vector μ is done through determination of the matrix. There are many components to be mixed up for the Gaussian distribution function and each has individual vector spaces in the distribution curve. The mixture distribution equation is expressed as:(27)$$P_{Q}\left(a\right)=\sum_{i=1}^{k}\propto_{j}.p\left(a|\mu_{j},\Sigma_{j}\right)$$

The jth Gaussian Mixture of the parameter is represented as $\mu_{j}\;\mathrm{and}\;\Sigma_{j}$ and, with the corresponding mixing coefficient, is represented as $\propto_{j}$.

5.Bayesian Linear Discriminant Classifier (BLDC)

The main usage of this type of classifier is to regularize the high-dimensional signal, the reduction of noisy signals, and to avoid the computation performance. An assumption to be made before proceeding to the Bayesian linear discriminant analysis, Zhou et al. [42], is that a target is set with respect to the relation in a vectors of b and c, which is denoted as white Gaussian noise; therefore, it is expressed as $a=x^{T}b+c$. The weighted function is considered as x, and its likelihood function is expressed as $p\left(G|\beta ,x\right)={\left(\frac{\beta }{2\pi }\right)}^{\frac{c}{2}}\exp\left(-\frac{\beta }{2}\left\|B^{T}x-m\right\|\right)$, where the pair of $\left\{B,m\right\}$ is denoted as G. The B matrix will give the training vector. a denotes the filtered signal, β denotes the inverse variance of the noise, and the sample size is denoted by C. The prior distribution of x is expressed as:(28)$$p\left(x|\alpha \right)={\left(\frac{\alpha }{2\pi }\right)}^{\frac{1}{2}}{\left(\frac{\varepsilon }{2\pi }\right)}^{\frac{1}{2}}\exp\left(-\frac{1}{2}x^{T}H^{\prime }\left(\alpha \right)x\right)$$

Here, the regularization square is represented as
$$H^{\prime }(\alpha )={\left[\begin{array}{ccc} \alpha & \cdots & 0\\ \vdots & \ddots & \vdots \\ 0 & \cdots & \varepsilon \end{array}\right]}_{\left(l+1)(l+1\right)}$$
where the hyper parameter α is produced from the forecasting the data, and the vector number is assigned as l. The weight x follows a Gaussian distribution which has zero mean, and a small value is contained in ε. According to the Bayes rule, the posterior distribution of x can be easily computed as:(29)$$p\left(x|\beta ,\alpha ,G\right)=\frac{P\left(G|\beta ,x\right)P\left(x|\alpha \right)}{\int P\left(G|\beta ,x\right)P\left(y|\alpha \right)dy}$$

For posterior distribution, the mean vector υ and the covariance matrix X should satisfy the norms in Equations (30) and (31). The nature of posterior distribution is highly Gaussian.
(30)$$\upsilon =\beta \;{\left(\beta BB^{T}+H^{\prime }\left(\alpha \right)\right)}^{-1}Ba$$
(31)$$X={\left(\beta BB^{T}+H^{\prime }\left(\alpha \right)\right)}^{-1}$$

For input prediction vector $\hat{b}$, the expression for the probability distribution on the regression is shown as $p\left(\hat{a}|\beta ,\alpha ,\hat{b,G}\right)=\int p\left(\hat{a}|\beta ,\hat{b,x}\right)p\left(x|\beta ,\alpha ,G\right)dy$.

The nature is, again, highly Gaussian in this prediction analysis, and its mean is expressed as $\mu =\upsilon^{T}\hat{b}$ while its variance is expressed as $\delta^{2}=\frac{1}{\beta }+\hat{b^{T}}X\hat{b}$.

6.Softmax Discriminant Classifier (SDC)

SDC [43] is included in this analysis for determination and identification of the group from which the specific test sample is taken. In this case, the weighing of its distance between the training samples to the test samples in a specific class or group of data is undertaken. The training set is denoted as:(32)$$Z=\left[Z_{1,}Z_{2},\ldots ,Z_{q}\right]\in R^{c\times d}$$
which comes from the distinct classes named q. $Z_{q}=\left[Z_{1}^{q},Z_{2}^{q},\ldots ,Z_{dq}^{q}\right]\in R^{c\times d_{q}},$ indicates the $d_{q}$ as samples from the $q^{th}$ class, where $\sum_{i=1}^{q}d_{i}=d.$ Assuming $K\in R^{c\times 1}$ is the test samples, and again it is given to the classifiers. If a negligible construction error can be obtained from the test samples, then we utilize the class of q. The class sample of q and test samples were transformed in the non-linear enhancing values, by which the ideology of SDC has been satisfied in the following equations:(33)$$h\left(K\right)=\arg\max Z_{w}^{i}$$
(34)$$h\left(K\right)=\arg\underset{i}{\max}\log\left(\sum_{j=1}^{d_{i}}\exp\left(-\lambda {\left\|v-\upsilon_{j}^{i}\right\|}_{2}\right)\right)$$

Here, $h\left(K\right)$ represents the distance between the ith class and the test samples. λ>0, validates the penalty cost. Hence, if K is identified as belonging to the class of the ith value, then $v\;and\;\upsilon_{j}^{i}$ are the same characteristic function, and so ${\left\|v-\upsilon_{j}^{i}\right\|}_{2}$ is improving close to zero and, hence, maximizing $Z_{w}^{i}$ can be achieved in the asymptotic values in which its maximum possibility is attained. 

7.Support Vector Machine– Radial Basis Function (SVM–RBF)

The SVM classifier is one of the important machine learning techniques for classification problems, especially the non-linear regression based multilayer perceptron with Radial Basis Function (RBF) by Yao, X. J., et al. [44].

The training time and computational complexity for the machine in the SVM depends on the classifier, and the data must be supported for the required SVMs. If we increase the support for the machines in the SVM, it leads to the computational requirements being high, and its floating point for the multiplication and addition is calculated.

Steps for the SVM is to identify

Step 1—With the help of quadratic optimization, we can use linearization and convergence. The dual optimization problem which was transformed from the primal minimization problem is referred to as maximizing the dual lagrangian *L_D_* with respect to $\alpha_{i}$,
(35)$$Max\;L_{D}=\sum_{i=1}^{l}\alpha_{i}-\frac{1}{2}\sum_{i=1}^{l}\sum_{j=1}^{l}\alpha_{i}\alpha_{j}y_{i}y_{j}(X_{i}\times X_{j})$$

Subject to $\sum_{i=1}^{l}\alpha_{i}y_{i}=0$, where $\alpha_{i}\geq 0\forall i=1,2,3,\ldots ,l$

Step 2—By solving the above quadratic programming problem in the optimal separating hyper plane, those points which have a non-zero lagrangian multiplier ($\propto_{i}\;>\;0)$ become the support vectors

Step 3—In the trained data, the optimal hyper plane is fixed by the support vectors, and it is very close to the decision boundary

Step 4—The K means clustering is the data set. It will function as a group of clusters according to the conditions of Step 2 and Step 3. A vector is randomly chosen from the clusters of three points each as a cluster or center point, which are the points from the given dataset. Each point in the center will acquire the present around them.

Step 5—If there are six center points from each corner, then the SVM training data are using kernel methods.
(36)$$\mathrm{Radial}\;\mathrm{Basis}\;\mathrm{Function}:\;k\;\left(x_{i},x_{j}\right)=\exp\left\{\frac{-{\left|x_{i}-x_{j}\right|}^{2}}{{\left(2*\sigma \right)}^{2}}\right\}$$

The hyper plane and support vectors are used to separate linearly separable and nonlinearly separable data.

## 7. Training and Testing of Classifiers

The training data for the dataset are limited. Therefore, we performed k-fold cross-validation. k-fold cross-validation is a popular method for estimating the performance of a machine learning model. The process performed by Fushiki et al. [45] for k-fold cross-validation is as follows. The first step is to divide the dataset into k equally sized subsets (or “folds”). For each fold *i*, the model is trained on all the data except the *i*th fold and the model is tested on the *i*th fold. The process is repeated for all k folds so that each is used once for testing. At the end of the process, you will have k performance estimates (one for each fold). Now, the average of the k performance estimates is calculated to get an overall estimate of the model’s performance. Once the model has been trained and validated using k-fold cross-validation, you can retrain it on the full dataset and predict new, unseen data. The advantage of k-fold cross-validation is that it provides a more reliable estimate of a model’s performance than a simple train–test split, as it uses all the available data. In this paper, the k-value is chosen as 10-fold. This research used a value of 2870 dimensionally reduced features per patient. This research is associated with 20 diabetic and 50 non-diabetic patients with multi-trail training of the required classifiers. The use of cross-validation removes any dependence on the choice of pattern for the test set. The training process is controlled by monitoring the Mean Square Error (MSE), which is defined as:(37)$$MSE=\frac{1}{N}\sum_{j=1}^{N}{\left(O_{j}-T_{j}\right)}^{2}$$
where *Oj*_-_ is the observed value at time *j*, *T_j_* is the target value at model *j*;

*j* = 1 and 2, and *N* is the total number of observations per epoch; in our case it is 2870. As the training progressed, the MSE value reached 1.0 × 10^−12^ within 2000 iterations.

Table 4 depicted the confusion matrix for Diabetic and Non-Diabetic Patient detection. 

In the case of diabetic detection, the following terms can be defined as:

True Positive (TP): A patient is correctly identified as diabetic class;

True Negative (TN): A patient is correctly identified as a non-diabetic class;

False Positive (FP): A patient is incorrectly identified as diabetic class when they are in non-diabetic class;

False Negative (FN): A patient is incorrectly identified as non-diabetic class when they are in diabetic class.

The training MSE always varied between 10^−4^ and 10^−8^, while the testing MSE varied from 10^−4^ to 10^−6^. The SVM (RBF) classifier without feature selection method settled at minimum training and testing MSEs of 1.26 × 10^−8^ and 5.141 × 10^−6^, respectively. The minimum testing MSE is one of the indicators towards the attainment of a better performance of the classifier. As shown in Table 5, a higher value of the testing MSE leads to a poorer performance of the classifier irrespective of the Dimensionality Reduction techniques.

Table 6 displays the training and testing MSE performance of the classifiers with the PSO feature selection method for four Dimensionality Reduction techniques. The training MSE always varied between 10^−5^ and 10^−8^, while the testing MSE varied from 10^−4^ to 10^−6^. The SVM (RBF) classifier with the PSO feature selection method settled at minimum training and testing MSEs of 1.94 × 10^−9^ and 1.885 × 10^−6^, respectively. All of the classifiers slightly improved the performance in the testing MSE when compared to the performance without feature selection methods. This will be indicated by the enhancement of the accuracy of the classifier performance irrespective of the type of the Dimensionality Reduction technique. 

Table 7 depicts the training and testing MSE performance of the classifiers with the Harmonic Search feature selection Method for four Dimensionality Reduction techniques. The training MSE always varied between 10^−5^ and 10^−8^, while the testing MSE varied from 10^−4^ to 10^−6^. The SVM (RBF) classifier with the Harmonic Search feature selection method settled at minimum training and testing MSEs of 1.86 × 10^−8^ and 1.7 × 10^−6^, respectively. All of the classifiers enhanced the performance in the testing MSE when compared to the performance without feature selection methods. This will be indicated by the improvement of the accuracy, MCC, and Kappa parameters of the classifier performance irrespective of the type of Dimensionality Reduction technique.

### Selection of Target

The target value for the non-diabetic case ${(T}_{ND})$ is taken at the lower side of the zero to one (0 → 1) scale and this mapping is made according to the constraint of:(38)$$\frac{1}{N}\sum_{i=1}^{N}\mu_{i}\leq T_{ND}$$
where $\mu_{i}$ is the mean value of input feature vectors for the *N* number of non-diabetic features taken for classification. Similarly, the target value for the diabetic cases ($T_{Dia}$) is taken at the upper side of the zero to one (0 → 1) scale and this mapping is made based on:(39)$$\frac{1}{M}\sum_{j=1}^{N}\mu_{j}\leq T_{Dia}$$
where $\mu_{j}$ is the average value of input feature vectors for the M number of diabetic cases taken for classification. Note that the target value $T_{Dia}$ would be greater than the average values of $\mu_{i}$ and $\mu_{j}$. The difference between the selected target values must be greater than or equal to 0.5, which is given by:(40)$$||T_{Dia}-T_{ND}||\geq 0.5$$

Based on the above constraints, the targets $T_{ND}$ and $T_{Dia}$ for the non-diabetic and diabetic patient output classes are chosen at 0.1 and 0.85, respectively. After selecting the target values, the Mean Squared Error (MSE) is used for evaluating the performance of machine learning classifiers.

Table 8 dictates the selection of optimum parametric values for classifiers. 

## 8. Results and Discussion

The research uses standard 10-fold testing and training, in which 10% of the input features are employed for testing, whereas 90% are employed for training. The choice of performance measures is significant in evaluating classifier performance. The confusion matrix is used to evaluate the performance of classifiers, especially in binary classification (i.e., classification into two classes, such as diabetic or non-diabetic from the pancreas microarray genes). It can be used to calculate performance metrics such as accuracy, F1 score, MCC, error rate, Jaccard metric, and Kappa, which are commonly used to evaluate the model’s overall performance. Table 9 depicts the parameters associated with the classifiers for performance analysis.

### 8.1. Performance Metrics

Accuracy

The accuracy of a classifier is a measure of how well it correctly identifies the class labels of a dataset. It is calculated by dividing the number of correctly classified instances by the total number of instances in the dataset. The equation for accuracy is given by Fawcett et al. [46]:$$Acc=\frac{\left(TN+TP\right)}{\left(TN+FN+TP+FP\right)}$$

2.Recall

Recall is a critical performance metric used to evaluate the classifier’s ability to correctly identify positive instances, specifically diabetic individuals, out of all the actual positive instances present in the dataset. It measures the proportion of true positive predictions out of all the instances that are positive in the dataset.
$$Recall=\frac{TP}{\left(TP+FN\right)}$$

3.Precision

Precision is used to evaluate the classifier’s ability to accurately predict positive instances. It measures the proportion of true positive predictions out of all instances predicted as positive by the classifier. A high precision value indicates that the classifier has a low false positive rate, meaning it correctly identifies a high proportion of diabetic individuals without misclassifying non-diabetic individuals as diabetic.
$$Precision=\frac{TP}{\left(TP+FP\right)}$$

4.F1 Score

The F1 score is a measure of a classifier’s accuracy that combines precision and recall into a single metric. It is calculated as the harmonic mean of precision and recall, with values ranging from 0 to 1, where 1 indicates perfect precision and recall. The equation for the F1 score is given by Saito et al. [47]:$$F1=\frac{2\times TP}{\left(2\times TP+FP+FN\right)}$$
where precision is the proportion of true positives among all instances classified as positive, and recall is the proportion of true positives among all instances that are positive. The F1 score is useful when the classes in the dataset are imbalanced, meaning there are more instances of one class than the other. In such cases, accuracy may not be a good metric to use, as a classifier that simply predicts the majority class would have a high accuracy but low precision and recall. The F1 score provides a more balanced measure of a classifier’s performance.

5.Matthews Correlation Coefficient (MCC)

MCC is a measure of the quality of binary (two-class) classification models. It considers true and false positives and negatives and is particularly useful in situations where the classes are imbalanced.

The MCC is defined by the following equation, as given in Chicco et al. [48]:$$MCC=\frac{\left(TP\times TN-FP\times FN\right)}{\sqrt{TP+FP)\times (TP+FN)\times (TN+FP)\times (TN+FN)}}$$

The MCC takes on values between −1 and 1, where a coefficient of 1 represents a perfect prediction, 0 represents a random prediction, and −1 represents a perfectly incorrect prediction.

6.Error Rate

The error rate of a classifier, as mentioned in Duda et al. [49], is the proportion of instances that are misclassified. It can be calculated using the following equation:$$Error\;rate=\frac{\left(FP+FN\right)}{\left(TP+TN+FP+FN\right)}$$

7.Jaccard Metric:

The Jaccard metric, also known as the Tanimoto similarity coefficient, explicitly disregards the accurate classification of negative samples [50].
$$Jaccard=\frac{TP}{TP+FP+FN}$$

Changes in data distributions can greatly impact the sensitivity of the Jaccard metric.

8.Kappa

The Kappa statistic, also known as Cohen’s Kappa, is a measure of agreement between two raters, or between a rater and a classifier. In the context of classification, it is used to evaluate the performance of a classifier on a binary or multi-class classification task. The Kappa statistic measures the agreement between the predicted and true classes, considering the possibility of agreement by chance. Kvålseth et al. [51] defined Kappa as follows:

$Kappa=considering\;P_{e}$ is the proportion of agreement expected by chance. $P_{o}$ and $P_{e}$ are calculated as follows:$$P_{o}=\frac{\left(TP+TN\right)}{\left(TP+TN+FP+FN\right)}$$$$P_{e}=\frac{\left(TP+FP)\times (TP+FN)+(FP+TN)\times (FN+TN\right)}{{\left(TP+TN+FP+FN\right)}^{2}}$$

The Kappa statistic takes on values between −1 and 1, where values greater than 0 indicate agreement better than chance, 0 indicates agreement by chance, and values less than 0 indicate agreement worse than chance. The results are tabulated in the following tables.

Table 9 demonstrates the performance analysis of the seven classifiers based on parameters like accuracy, F1 score, MCC, error rate, Jaccard metric, and Kappa values for the four Dimensionality Reduction methods without feature selection methods. It can be identified from Table 9 that the SVM (RBF) classifier in the Cuckoo Search DR technique is settled at a middle accuracy of 65.71%, F1 score of 50% with a moderate error rate of 34.28%, and Jaccard metric of 33.33%. The SVM (RBF) classifier also exhibits a low value of MCC 0.2581 and Kappa value of 0.25. The Logistic Regression classifier for the firefly algorithm DR Technique is placed in the lower ebb of accuracy of 51.42%, with a high error rate of 48.57%, F1 score of 37.03%, and Jaccard metric of 22.72%. The MCC and Kappa values of the Logistic Regression classifier are 0.01807 and 0.01652, respectively. Irrespective of the Dimensionality Reduction techniques, all of the classifiers settled at an accuracy within the range of 50–65%. This is due to the inherent limitation of the Dimensionality Reduction techniques. Therefore, it is recommended to incorporate the feature selection methods to enhance the classifier performance.

Figure 9 depicts the performance analysis of the seven classifiers based on parameters such as accuracy, F1 score, error rate, and Jaccard metric values for the four Dimensionality Reduction methods without feature selection methods. These attained a middle accuracy of 65.71%, F1 score of 50% with a moderate error rate of 34.28%, and Jaccard metric of 33.33%. The Logistic Regression classifier for the firefly algorithm DR Technique was placed in the lower end of accuracy, with a value of 51.42%, a high error rate of 48.57%, an F1 score of 37.03%, and a Jaccard metric of 22.72%. 

Table 10 exhibits the performance analysis of the seven classifiers for the four Dimensionality Reduction methods with the PSO feature selection method. It can be observed from Table 10 that the SVM (RBF) classifier in the Chi square pdf DR techniques is settled at a high accuracy of 91.42%, F1 score of 85.71% with a low error rate of 8.57%, and Jaccard metric of 75%. The SVM (RBF) classifier also exhibits a high value of MCC 0.7979 and Kappa value of 0.7961. The Logistic Regression classifier for the firefly algorithm DR technique is once again placed in the lower end of accuracy, with a value of 55.71%, with a high error rate of 44.28%, F1 score of 43.63%, and Jaccard metric of 27.9%. The MCC and Kappa values of the Logistic Regression classifier are 0.1264 and 0.1142, respectively. Irrespective of the Dimensionality Reduction techniques, all of the classifiers settled at an accuracy within the range of 55–92%. This enhancement in the accuracy is due to the inherit property of the PSO feature selection method.

Figure 10 displays the performance analysis of the seven classifiers for the four Dimensionality Reduction methods with PSO feature selection methods. It is also identified from Figure 10 that the SVM (RBF) classifier in the Chi Square pdf DR techniques is settled at a high accuracy of 91.42%, F1 score of 85.71% with low an error rate of 8.57%, and Jaccard metric of 75%. The Logistic Regression classifier for the firefly algorithm DR technique settled in the lower end of accuracy with a value of 55.71%, with a high error rate of 44.28% F1 score of 43.63%, and Jaccard metric of 27.9%. The PSO feature selection method improves the classifier accuracy by around 10–35%, irrespective of the DR techniques.

Table 11 explores the performance analysis of the seven classifiers for the four Dimensionality Reduction methods with the Harmonic Search feature selection method. It is observed from Table 11 that the SVM (RBF) classifier in the Cuckoo search DR techniques is settled at a high accuracy of 90%, F1 score of 83.72% with a low error rate of 10%, and Jaccard metric of 72%. The SVM (RBF) Classifier also exhibits a high value of MCC 0.7694 and Kappa value of 0.7655. The Linear Regression classifier for the Detrend Fluctuation Analysis (DFA) DR technique is placed in the lower accuracy of 52.85%, with a high Error Rate of 47.14%, F1 score of 37.75%, and Jaccard metric of 23.25%. The MCC and Kappa values of the Linear Regression classifier are 0.0361 and 0.03343, respectively. Irrespective of the Dimensionality Reduction techniques, all of the classifiers settled at an accuracy within the range of 50–90%. This enhancement in the accuracy is due to the usage of the Harmonic Search Feature selection method.

Figure 11 exhibits the performance analysis of the seven classifiers for the four Dimensionality Reduction methods with the Harmonic Search feature selection methods. It is also observed from Figure 11 that the SVM (RBF) classifier in the Cuckoo search DR techniques is settled at a high accuracy of 90%, F1 score of 83.72% with a low error rate of 10%, and Jaccard metric of 72%. The Linear Regression classifier for the Detrend Fluctuation Analysis (DFA) DR technique is settled in the lower accuracy of 52.85%, with a high error rate of 47.14%, F1 score of 37.75%, and Jaccard metric of 23.25%. The Harmonic Search feature selection method improves the classifier accuracy by around 10–25%, irrespective of the DR techniques, and achieved the position next to the PSO feature selection method.

Figure 12 displays the performance of the MCC and Kappa parameters across the classifier for the four DR Techniques without and with the two-feature selection methods. The MCC and Kappa are the benchmark parameters which indicate the outcomes of the classifiers for different inputs. As in this research, there are three categories of inputs, like dimensionally reduced without feature selection, with PSO and Harmonic Feature selection methods. The classifiers’ performances are observed through the attained MCC and Kappa values for these inputs. The average MCC and Kappa values from the classifiers are 0.2984 and 0.2849, respectively. A methodology was devised to identify the performance of the classifiers with reference to Figure 12. The MCC values are divided into three ranges, 0.01–0.25, 0.251–0.54, and 0.55–0.8. The performance of the classifiers is very poor in the range 1 and there is a steep increase in the MCC vs Kappa slope in region 2 of the MCC values. Region 3 of the MCC values settled at a higher performance of the classifiers without any glitches.

### 8.2. Computational Complexity

The classifiers are analysed based on the computational complexity. The computational complexity is found according to the input O(n) size. The computational complexity is less if it equals O(1). As the number of inputs increases, the computational complexity will increase. In this research, the complexity does not depend on the input size; this is one of the desired entities for any algorithm. If the computational complexity increases log(n) times for any increase in ‘n,’ it is denoted as O(logn). In this paper, all of the classifiers are hybrid, and they classify the dimensionally reduced outputs along with the feature selection methods.

Table 12 shows the computational complexity of the classifiers for the four Dimensionality Reduction techniques without feature selection methods. It is observed from Table 12 that almost all of the classifiers’ computational complexities are near equal, and their performances are positioned in the low level of accuracy. The Linear Regression classifier has a low computational complexity of O(2nlog2n); at the same time, the Logistic Regression classifier with the Firefly algorithm DR techniques has a higher complexity of O(2n^3^log2n) and both of the classifiers are at the same level of accuracy. The SVM (RBF) classifier for the Cuckoo search DR technique has a high computational complexity of O(2n^3^log4n) with increased accuracy, MCC, and Kappa values.

Table 13 displays the computational complexity of the classifiers for the four Dimensionality Reduction techniques with the PSO feature selection method. It is identified from Table 13 that almost all of the classifiers’ computational complexities are near equal, and their performances are positioned in the high level of accuracy. The Linear Regression classifier has a low computational complexity of O(2n^3^log2n); at the same time, the Logistic Regression classifier for the Firefly algorithm DR techniques has a higher complexity of O(2n^4^log2n), and both the classifiers are at the same level of accuracy. The SVM (RBF) classifier for the Chi square pdf DR technique has a high computational complexity of O(2n^4^log4n), with the highest accuracy of 91%, and MCC and Kappa values of 0.794 and 0.7967, respectively.

Table 14 shows the computational complexity of the classifiers for the four Dimensionality Reduction techniques with the Harmonic Search feature selection method. It can be seen in Table 14 that almost all of the classifiers’ computational complexities are near equivalent and their performances are positioned in the high level of accuracy. The Linear Regression classifier has a low computational complexity of O(2n^2^log2n); at the same time, the BDLC and GMM classifiers with the Firefly algorithm DR techniques have a higher complexity of O (2n^5^log2n), and both the classifiers are at the same level of accuracy. The SVM (RBF) classifier for the Cuckoo search DR technique has a high computational complexity of O(2n^4^log2n), with the highest accuracy of 90%, and MCC and Kappa values of 0.7655 and 0.767, respectively. Even though there is a high computational complexity associated with the GMM and BDLC classifiers, they have not achieved better performance metrics.

### 8.3. Comparison of Previous Works

Table 15 shows the comparision of previous works with different machine learning techniques to detect the diabetics.

As mentioned in the Table 15 it is observed that, majority of the machine learning classifiers such as SVM (RBF), Naïve Bayes, Logistic Regression, Decision tree, Non-linear regression, random forest, multilayer perceptron, and Deep neural networks are utilized to classify the diabetics, based on the clinical data base. All the classifiers accuracy is at the range of 67–91%. The current study is based on microarray gene to detect diabetes and SVM (RBF) achieved accuracy of 91%. 

Table 16 explains the comparison of classifiers performance in different data sets. It is identify that, from Table 16, explores the efficacy of machine learning classifiers in detecting various diseases with different datasets. It is also noted that, the clinical diabetes dataset from the Frankfurt hospital, Germany using decision tree (ID3) classifier attains highest accuracy of 99%. 

## 9. Conclusions

The main aim of this paper is to identify and analyze a good classifier to classify diabetic microarray gene data with a high accuracy and low error rate. To assimilate the theme performances by seven classifiers, the classifiers are analyzed through benchmark measures like accuracy, F1 score, MCC, error rate, Jaccard metric, and Kappa. The highest accuracy of 91% is achieved in the SVM (RBF) classifier for the Chi^2^ pdf DR technique with the PSO feature selection method. The classifier exhibited a high computational complexity of O(2n^4^ log 2n). The second highest accuracy of 90% is achieved through the SVM (RBF) classifier, in combination with the Cuckoo search DR technique with the Harmonic search feature selection method; the computational complexity is O (2n^4^ log2n). However, even the GMM and BLDC classifiers with higher computational complexity failed to achieve good accuracy. The enhanced accuracy attained by the SVM (RBF) classifier will be utilized for better classification of diabetic microarray gene data. Future research will be in the direction of CNN and DNN models for the improvement of classifier performance.

## Figures and Tables

**Figure 1 diagnostics-13-02654-f001:**
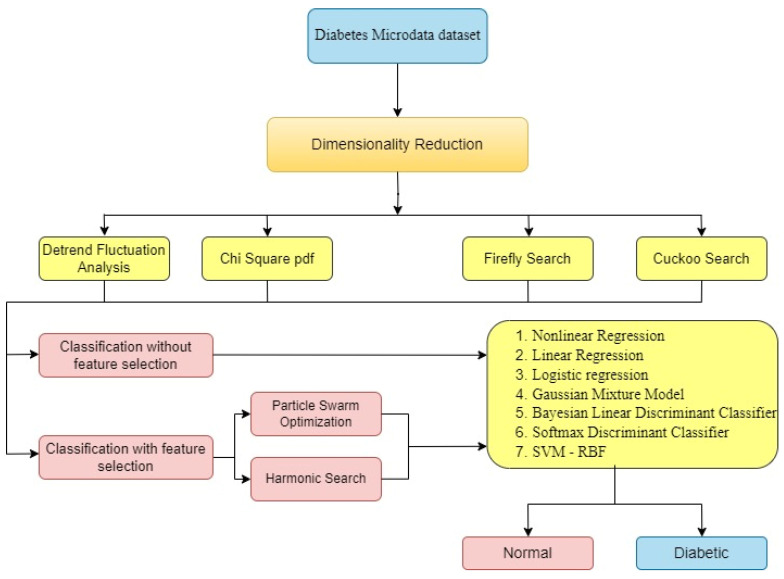
Illustration of the work.

**Figure 2 diagnostics-13-02654-f002:**
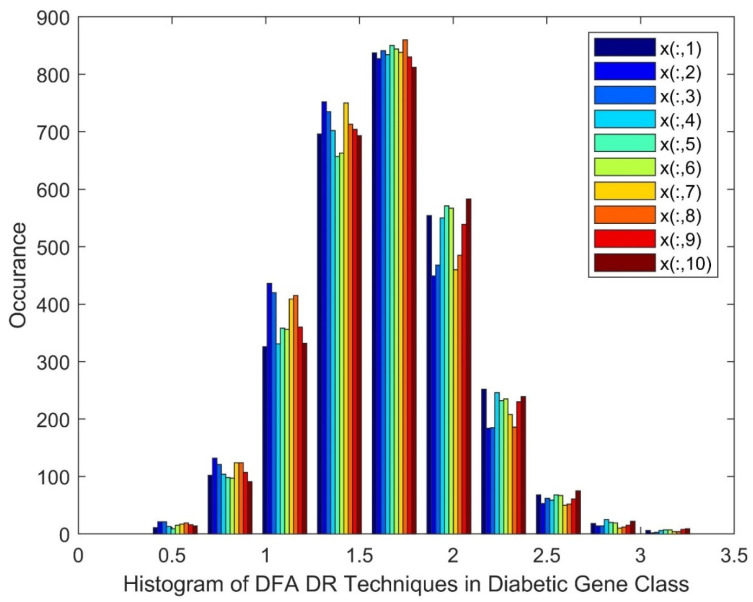
Histogram of Detrend Fluctuation Analysis (DFA) Techniques in Diabetic Gene Class.

**Figure 3 diagnostics-13-02654-f003:**
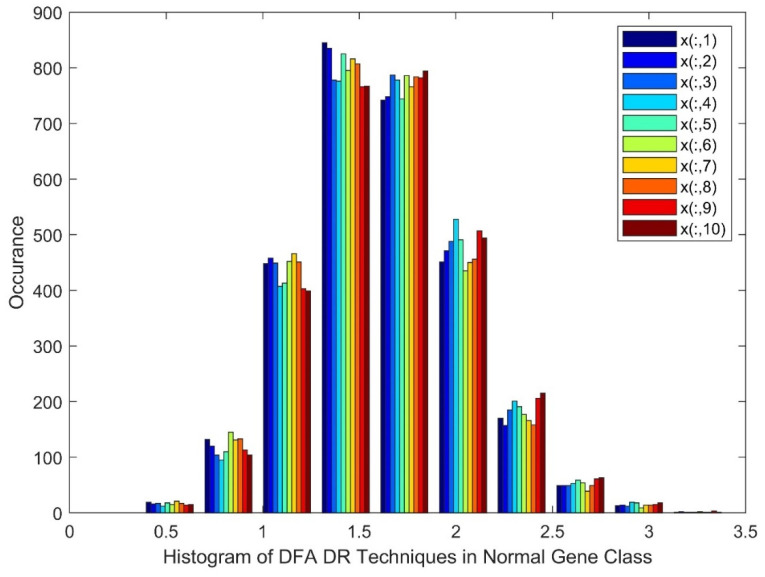
Histogram of Detrend Fluctuation Analysis (DFA) Techniques in Normal Gene Class.

**Figure 4 diagnostics-13-02654-f004:**
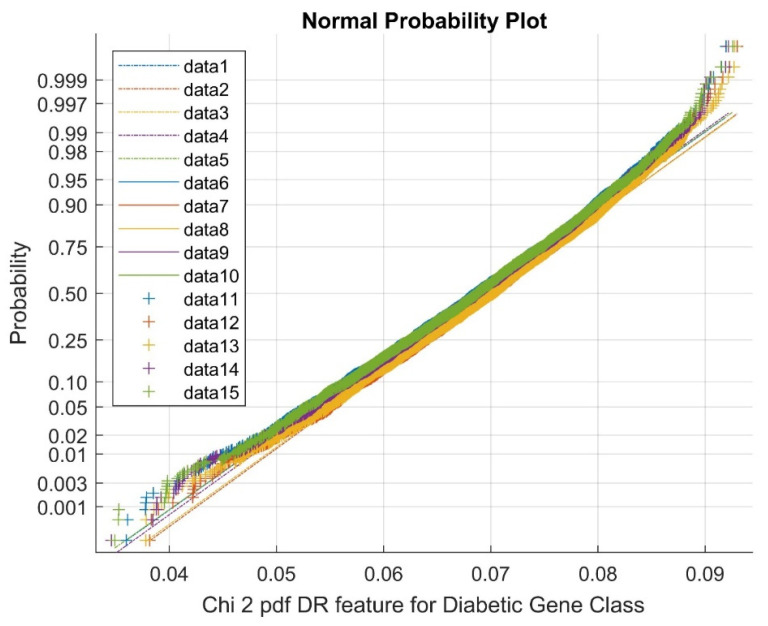
Normal Probability plot for Chi Square DR Techniques Features for Diabetic Gene Class.

**Figure 5 diagnostics-13-02654-f005:**
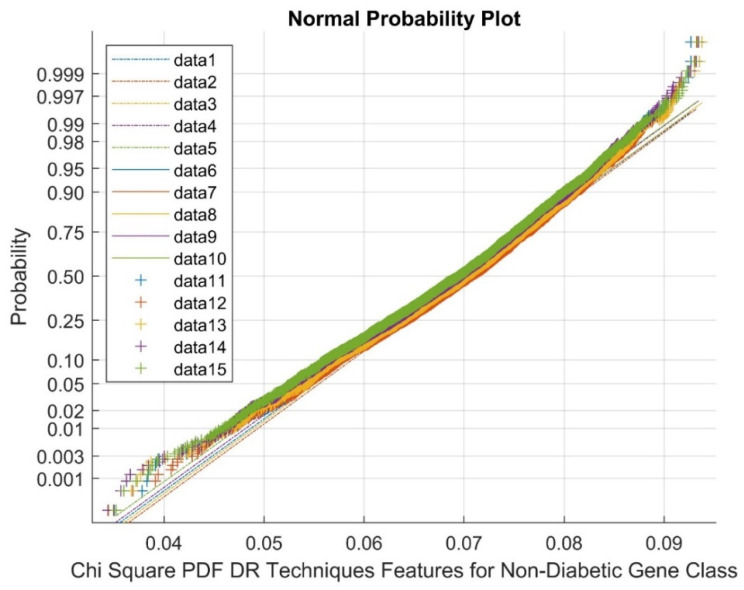
Normal Probability plot for Chi Square PDF DR Techniques Features for Non-Diabetic Gene Class.

**Figure 6 diagnostics-13-02654-f006:**
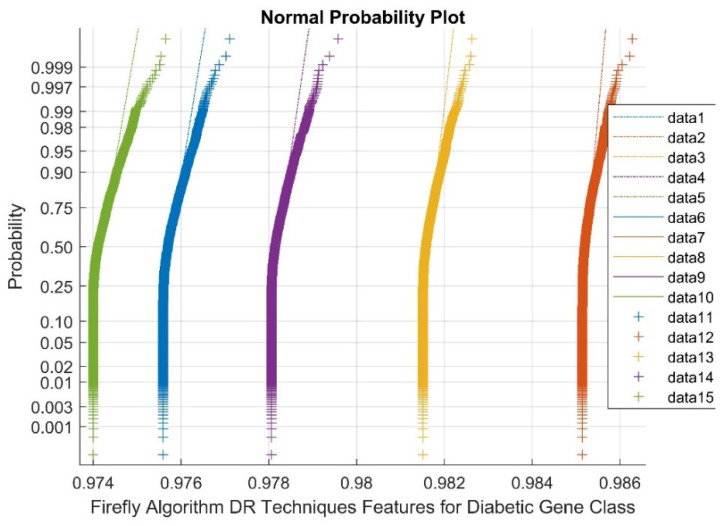
Normal Probability plot for Firefly Algorithm DR Techniques Features for Diabetic Gene Class.

**Figure 7 diagnostics-13-02654-f007:**
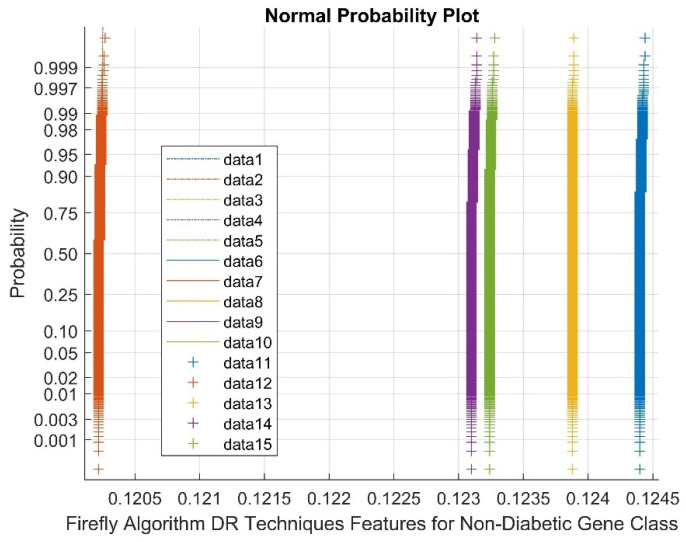
Normal Probability plot for Firefly algorithm DR techniques features for non-diabetic gene class.

**Figure 8 diagnostics-13-02654-f008:**
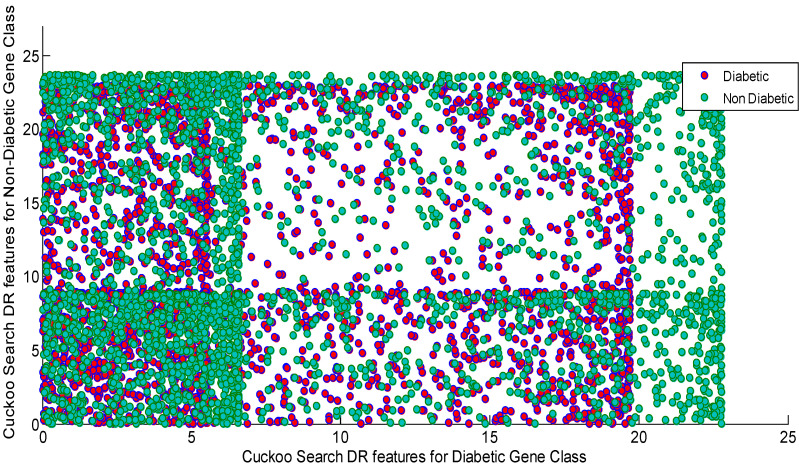
Scatter plot for Cuckoo search DR technique with diabetic and non-diabetic gene classes.

**Figure 9 diagnostics-13-02654-f009:**
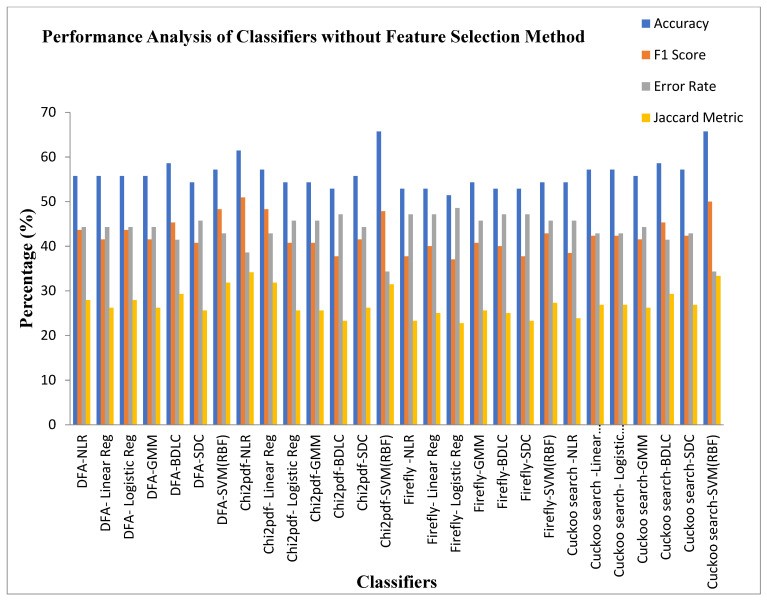
Performance Analysis of Classifiers without Feature Selection Methods.

**Figure 10 diagnostics-13-02654-f010:**
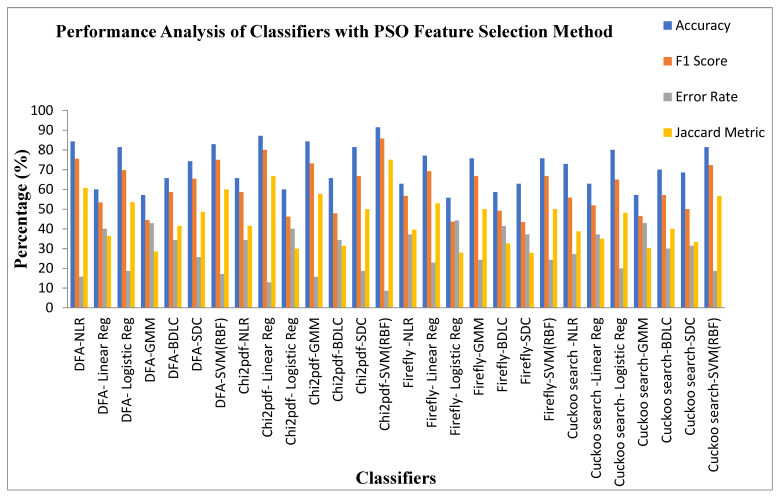
Performance Analysis of Classifiers with PSO Feature Selection Methods.

**Figure 11 diagnostics-13-02654-f011:**
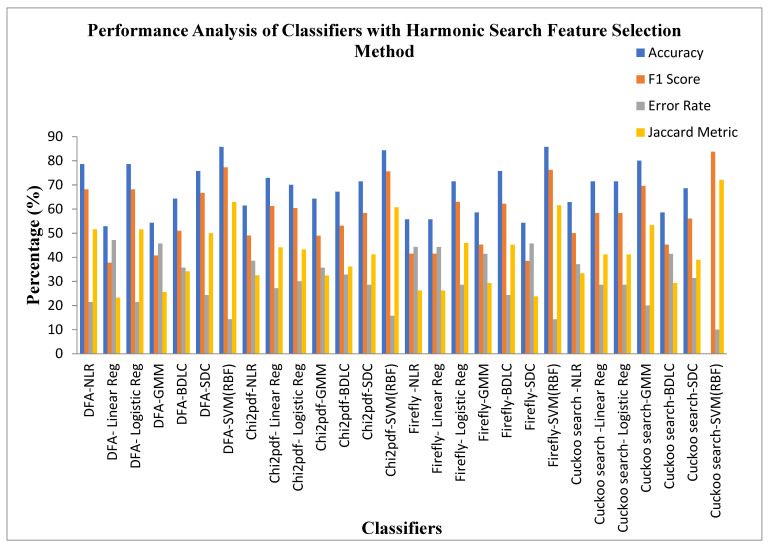
Performance analysis of classifiers with Harmonic Search feature selection methods.

**Figure 12 diagnostics-13-02654-f012:**
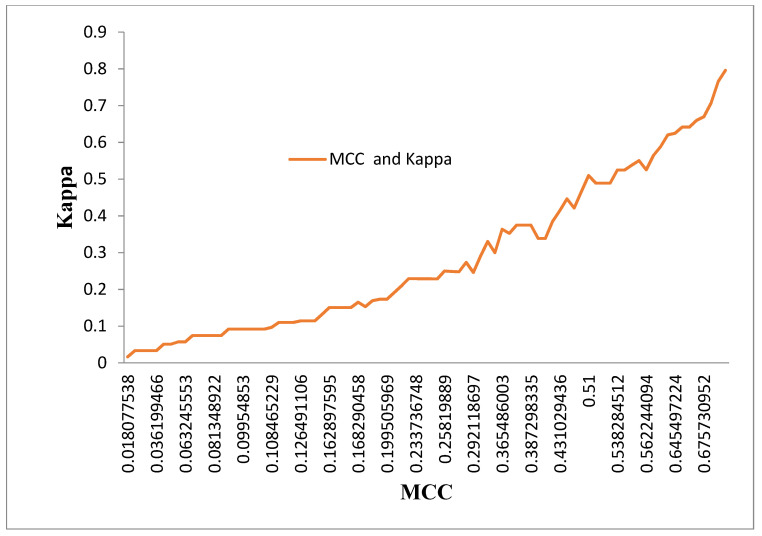
Performance of MCC and Kappa parameters across the classifier for four DR techniques without and with two-feature selection methods.

**Table 1 diagnostics-13-02654-t001:** Description of Pancreas Microarray Gene Data set for Diabetic and Non-Diabetic Classes.

Data Set	No of Genes	Class 1 Diabetes	Class 2 Non-Diabetic	Total
Pancreas	28735	20	50	70

**Table 2 diagnostics-13-02654-t002:** Statistical Analysis for Different Dimensionality Reduction Techniques.

Statistical Parameters	DFA		Chi^2^pdf		Firefly Algorithm		Cuckoo Search	
	DP	NDP	DP	NDP	DP	NDP	DP	NDP
Mean	1.6302	1.6301	0.0690	0.0691	1.0278	0.1260	8.7158	12.6033
Variance	0.1614	0.1665	0.0007	0.0008	0.0004	1.45 × 10^−9^	48.4751	71.7672
Skewness	0.2319	0.2758	−0.2771	−0.3228	1.6527	4.0298	0.6087	0.2692
Kurtosis	0.2706	0.1600	0.0301	0.0096	3.1033	78.3911	−1.1233	−1.3759
Pearson CC	0.9516	0.9598	0.9781	0.9814	0.8803	0.5006	0.7048	0.6859
CCA	**0.05914**		**0.06411**		**0.04785**		**0.05107**	

**Table 3 diagnostics-13-02654-t003:** *p*-value significant for Feature Selection method from *t*-test for different DR techniques.

Feature Selection	DR Techniques	DFA		Chi Square pdf		Firefly Algorithm		Cuckoo Search	
	Class	DP	NDP	DP	NDP	DP	NDP	DP	NDP
**PSO**	*p* value < 0.05	0.415	0.1906	0.3877	0.16074	0.38059	0.2435	0.4740	0.48824
**Harmonic Search**	*p* value < 0.05	**0.0004**	0.4290	0.3836	0.43655	**0.00031**	0.469	0.3488	0.3979

**Table 4 diagnostics-13-02654-t004:** Confusion Matrix for Diabetic and Non-Diabetic Patient Detection.

Truth of Clinical Situation		Predicted Values	
		Diabetic	Non-Diabetic
Actual Values	Diabetic	TP	FN
	Non-Diabetic	FP	TN

**Table 5 diagnostics-13-02654-t005:** Training and Testing MSE Performance of the Classifiers Without Feature Selection Method for Four Dimensionality Reduction Techniques.

Classifiers	With DFA DR Method		With Chi^2^ pdf DR Method		With Firefly Algorithm DR Method		With Cuckoo Search DR Method	
	Training MSE	Testing MSE	Training MSE	Testing MSE	Training MSE	Testing MSE	Training MSE	Testing MSE
NLR	7.92 × 10^−5^	1.633 × 10^−4^	9.8 × 10^−5^	5.439 × 10^−5^	6.56 × 10^−5^	2.9 × 10^−4^	2.81 × 10^−5^	1.45 × 10^−4^
Linear Reg	4.62 × 10^−5^	8.936 × 10^−5^	6.08 × 10^−5^	1.945 × 10^−4^	5.48 × 10^−5^	3.566 × 10^−4^	9.22 × 10^−5^	1.35 × 10^−4^
Logistic Regression	2.6 × 10^−5^	1.325 × 10^−4^	2.21 × 10^−5^	1.565 × 10^−4^	8.84 × 10^−5^	4.625 × 10^−4^	9.8 × 10^−5^	1.685 × 10^−4^
GMM	1.52 × 10^−5^	1.21 × 10^−4^	7.06 × 10^−5^	1.662 × 10^−4^	7.92 × 10^−5^	2.442 × 10^−4^	3.42 × 10^−5^	1.103 × 10^−4^
BDLC	1.82 × 10^−6^	6.978 × 10^−5^	8.46 × 10^−5^	2.492 × 10^−4^	4.49 × 10^−5^	2.136 × 10^−4^	1.44 × 10^−5^	7.161 × 10^−5^
SDC	2.72 × 10^−5^	1.567 × 10^−4^	1.21 × 10^−5^	1.095 × 10^−4^	4.22 × 10^−5^	3.081 × 10^−4^	1.21 × 10^−5^	1.04 × 10^−4^
SVM (RBF)	1.96 × 10^−5^	1.656 × 10^−4^	4.62 × 10^−5^	1.454 × 10^−4^	3.48 × 10^−5^	1.924 × 10^−4^	1.26 × 10^−8^	**5.141** × 10^−6^

**Table 6 diagnostics-13-02654-t006:** Training and Testing MSE Analysis of Classifiers for Various Dimensionality Reduction Techniques with PSO Feature Selection Methods.

Classifiers	With DFA DR Method		With Chi^2^ pdf DR Method		With Firefly Algorithm DR Method		With Cuckoo Search DR Method	
	Training MSE	Testing MSE	Training MSE	Testing MSE	Training MSE	Testing MSE	Training MSE	Testing MSE
NLR	3.61 × 10^−6^	9.01 × 10^−6^	5.29 × 10^−6^	4.062 × 10^−5^	3.36 × 10^−6^	1.568 × 10^−4^	3.84 × 10^−7^	3.737 × 10^−5^
Linear Reg	9.61 × 10^−6^	1.603 × 10^−5^	1.22 × 10^−7^	4.825 × 10^−6^	8.28 × 10^−6^	1.464 × 10^−5^	4.62 × 10^−6^	5.042 × 10^−5^
Logistic Regression	1.44 × 10^−5^	1.422 × 10^−6^	2.6 × 10^−8^	6.806 × 10^−6^	6.89 × 10^−5^	1.486 × 10^−4^	7.4 × 10^−7^	2.906 × 10^−5^
GMM	6.24 × 10^−5^	8.021 × 10^−5^	1.36 × 10^−7^	1.168 × 10^−5^	4.62 × 10^−6^	1.394 × 10^−5^	2.7 × 10^−7^	1.511 × 10^−4^
BDLC	1.76 × 10^−5^	4.063 × 10^−5^	7.29 × 10^−6^	5.653 × 10^−5^	1.52 × 10^−6^	1.192 × 10^−4^	5.33 × 10^−7^	2.92 × 10^−5^
SDC	9 × 10^−6^	2.003 × 10^−5^	2.89 × 10^−6^	2.493 × 10^−5^	1.02 × 10^−5^	1.243 × 10^−4^	8.65 × 10^−8^	5.525 × 10^−5^
SVM (RBF)	2.12 × 10^−7^	8.5 × 10^−6^	1.94 × 10^−9^	**1.885** × 10^−6^	2.56 × 10^−6^	1.889 × 10^−5^	7.22 × 10^−7^	1.022 × 10^−5^

**Table 7 diagnostics-13-02654-t007:** Training and Testing MSE Analysis of Classifiers for Various Dimensionality Reduction Technique with Harmonic Search Feature Selection Methods.

Classifiers	With DFA DR Method		With Chi^2^ pdf DR Method		With Firefly Algorithm DR Method		With Cuckoo Search DR Method	
	Training MSE	Testing MSE	Training MSE	Testing MSE	Training MSE	Testing MSE	Training MSE	Testing MSE
NLR	7.06 × 10^−5^	1.68 × 10^−5^	3.25 × 10^−5^	5.46 × 10^−5^	5.04 × 10^−5^	9.7 × 10^−5^	9.61 × 10^−6^	5.07 × 10^−5^
Linear Reg	9.22 × 10^−5^	2.44 × 10^−4^	1.44 × 10^−5^	2.22 × 10^−5^	1.68 × 10^−5^	1.21 × 10^−4^	8.28 × 10^−7^	2.86 × 10^−5^
Logistic Regression	5.93 × 10^−8^	1.68 × 10^−5^	6.25 × 10^−6^	3.03 × 10^−5^	2.81 × 10^−7^	2.86 × 10^−5^	3.25 × 10^−5^	3.14 × 10^−5^
GMM	1.09 × 10^−5^	2.21 × 10^−4^	1.76 × 10^−5^	5.66 × 10^−5^	3.02 × 10^−5^	7.57 × 10^−5^	3.84 × 10^−5^	1.23 × 10^−5^
BDLC	2.92 × 10^−8^	5.04 × 10^−5^	1.82 × 10^−4^	4.22 × 10^−5^	3.25 × 10^−5^	2.58 × 10^−5^	1.85 × 10^−5^	7.42 × 10^−5^
SDC	6.56 × 10^−8^	1.72 × 10^−5^	1.69 × 10^−4^	2.5 × 10^−5^	5.76 × 10^−6^	2.25 × 10^−4^	1.52 × 10^−5^	3.72 × 10^−5^
SVM (RBF)	4.36 × 10^−5^	4.88 × 10^−6^	7.4 × 10^−5^	8.13 × 10^−6^	6.56 × 10^−5^	1.02 × 10^−5^	1.86 × 10^−7^	**1.7** × 10^−6^

**Table 8 diagnostics-13-02654-t008:** Selection of Optimum Parametric Values for Classifiers.

Classifiers	Description
NLR	Set distribution as N(o,σ^2^), g < 0.2, H < 0.014 with τ = 1, Convergence Criteria: MSE
LR	$\theta^{T}$< 0.6, β = 0.01 and Convergence Criteria: MSE
LoR	Threshold H θ(x) = 0.5. Criterion: MSE
GMM	Mean, covariance of the input samples and tuning parameter as like in Expectation Maximum in test point likelihood probability 0.15, cluster probability of 0.6, with a convergence rate of 0.6. Criterion: MSE
BLDC	Prior probability P(x): 0.5, class mean µx = 0.8 and µy = 0.1. Criterion: MSE
SDC	C: 0.5, Coefficient of the kernel function (gamma): 10, Class weights: 0.5, Convergence Criteria: MSE
SVM–RBF	C: 1, Coefficient of the kernel function (gamma): 100, Class weights: 0.86, Convergence Criteria: MSE

**Table 9 diagnostics-13-02654-t009:** Average Performance of Classifiers with Different DM Techniques Without Feature Selection Methods.

Dimensionality Reduction	Classifiers	Parameters							
		Accuracy (%)	Recall	Precision	F1Score (%)	MCC	Error Rate (%)	Jaccard Metric (%)	Kappa
Detrend Fluctuation Analysis (DFA)	NLR	55.7142	60	34.2857	43.6363	0.1264	44.2857	27.9069	0.1142
	Linear Reg	55.7142	55	33.3333	41.5094	0.0995	44.2857	26.1904	0.0920
	Logistic Regression	55.7142	60	34.2857	43.6363	0.1264	44.2857	27.9069	0.1142
	GMM	55.7142	55	33.3333	41.5094	0.0995	44.2857	26.1904	0.0920
	BDLC	58.5714	60	36.3636	45.2830	0.1628	41.4285	29.2682	0.1506
	SDC	54.2857	55	32.3529	40.7407	0.0813	45.7142	25.5813	0.0743
	SVM (RBF)	57.1428	70	36.8421	48.2758	0.1995	42.8571	31.8181	0.1732
Chi^2^pdf	NLR	61.4285	70	40	50.909	0.2529	38.5714	34.1463	0.2285
	Linear Reg	57.1428	70	36.8421	48.2758	0.1995	42.8571	31.8181	0.1732
	Logistic Regression	54.2857	55	32.3529	40.7407	0.0813	45.7142	25.5813	0.0743
	GMM	54.2857	55	32.3529	40.7407	0.0813	45.7142	25.5813	0.0743
	BDLC	52.8571	50	30.3030	37.7358	0.0361	47.1428	23.2558	0.0334
	SDC	55.7142	55	33.3333	41.5094	0.0995	44.2857	26.1904	0.0920
	SVM (RBF)	65.7142	55	42.3076	47.826	0.2337	34.2857	31.4285	0.2293
Firefly Algorithm	NLR	52.8571	50	30.3030	37.7358	0.0361	47.1428	23.2558	0.0334
	Linear Reg	52.8571	55	31.4285	40	0.0632	47.1428	25	0.0571
	Logistic Regression	51.4285	50	29.4117	37.037	0.0180	48.5714	22.7272	0.0165
	GMM	54.2857	55	32.3529	40.7407	0.0813	45.7142	25.5813	0.0743
	BDLC	52.8571	55	31.4285	40	0.0632	47.1428	25	0.0571
	SDC	52.8571	50	30.3030	37.7358	0.0361	47.1428	23.2558	0.0334
	SVM (RBF)	54.2857	60	33.3333	42.8571	0.1084	45.7142	27.2727	0.0967
Cuckoo Search	NLR	54.2857	50	31.25	38.4615	0.0544	45.7142	23.8095	0.0508
	Linear Reg	57.1428	55	34.375	42.3076	0.1178	42.8571	26.8292	0.1101
	Logistic Regression	57.1428	55	34.375	42.3076	0.1178	42.8571	26.8292	0.1101
	GMM	55.7142	55	33.3333	41.5094	0.0995	44.2857	26.1904	0.0920
	BDLC	58.5714	60	36.3636	45.2830	0.1628	41.4285	29.2682	0.1506
	SDC	57.1428	55	34.375	42.3076	0.1178	42.8571	26.8292	0.1101
	SVM (RBF)	**65.7142**	**60**	**42.8571**	**50**	**0.2581**	**34.2857**	**33.3333**	**0.25**

**Table 10 diagnostics-13-02654-t010:** Average Performance of Classifiers with Different DM Techniques With PSO Feature Selection Method.

Dimensionality Reduction	Classifiers	Parameters							
		Accuracy (%)	Recall	Precision	F1 Score (%)	MCC	Error Rate (%)	Jaccard Metric (%)	Kappa
Detrend Fluctuation Analysis (DFA)	NLR	84.2857	85	68	75.5555	0.6505	15.7142	60.7142	0.6418
	Linear Reg	60	80	40	53.3333	0.2921	40	36.3636	0.2461
	Logistic Regression	81.4285	75	65.2173	69.7674	0.5674	18.5714	53.5714	0.5645
	GMM	57.1428	60	35.2941	44.4444	0.1446	42.8571	28.5714	0.1322
	BDLC	65.7142	85	44.7368	58.6206	0.3899	34.2857	41.4634	0.3385
	SDC	74.2857	85	53.125	65.3846	0.4987	25.7142	48.5714	0.4661
	SVM (RBF)	82.8571	90	64.2857	75	0.6454	17.1428	60	0.625
Chi^2^pdf	NLR	65.7142	85	44.73684	58.6206	0.3899	34.2857	41.4634	0.3385
	Linear Reg	87.1428	90	72	80	0.7165	12.8571	66.6666	0.7069
	Logistic Regression	60	60	37.5	46.1538	0.1813	40	30	0.1694
	GMM	84.2857	75	71.4285	73.1707	0.6210	15.7142	57.6923	0.6206
	BDLC	65.7142	55	42.3076	47.8260	0.2337	34.2857	31.4285	0.2293
	SDC	81.4285	65	68.4210	66.6666	0.5384	18.5714	50	0.5380
	SVM (RBF)	**91.4285**	**90**	**81.8181**	**85.7142**	**0.7979**	**8.57142**	**75**	**0.7961**
Firefly Algorithm	NLR	62.8571	85	42.5	56.6666	0.3560	37.1428	39.5348	0.3
	Linear Reg	77.1428	90	56.25	69.2307	0.5622	22.8571	52.9411	0.5254
	Logistic Regression	55.7142	60	34.2857	43.6363	0.1264	44.2857	27.9069	0.1142
	GMM	75.7142	85	54.8387	66.6666	0.5183	24.2857	50	0.4892
	BDLC	58.5714	70	37.8378	49.1228	0.2171	41.4285	32.5581	0.1912
	SDC	62.8571	50	38.4615	43.4782	0.1682	37.1428	27.7777	0.1651
	SVM (RBF)	75.7142	85	54.8387	66.6666	0.5183	24.2857	50	0.4892
Cuckoo Search	NLR	72.8571	60	52.17391	55.8139	0.3654	27.1428	38.7096	0.3636
	Linear Reg	62.8571	70	41.17647	51.8518	0.2711	37.1428	35	0.2479
	Logistic Regression	80	65	65	65	0.51	20	48.1481	0.51
	GMM	57.1428	65	36.1111	46.4285	0.1717	42.8571	30.2325	0.1532
	BDLC	70	70	48.2758	57.1428	0.3668	30	40	0.3524
	SDC	68.5714	55	45.8333	50	0.2760	31.4285	33.3333	0.2735
	SVM (RBF)	81.4285	85	62.9629	72.3404	0.6032	18.5714	56.6666	0.5882

**Table 11 diagnostics-13-02654-t011:** Average Performance of Classifiers with Different DM Techniques with Harmonic Search Feature Selection Method.

Dimensionality Reduction	Classifiers	Parameters							
		Accuracy (%)	Recall	Precision	F1 Score (%)	MCC	Error Rate (%)	Jaccard Metric (%)	Kappa
Detrend Fluctuation Analysis (DFA)	NLR	78.5714	80	59.2592	68.0851	0.5382	21.4285	51.6129	0.5248
	Linear Reg	52.8571	50	30.3030	37.7358	0.0361	47.1428	23.2558	0.0334
	Logistic Regression	78.5714	80	59.2592	68.0851	0.5382	21.4285	51.6129	0.5248
	GMM	54.2857	55	32.3529	40.7407	0.0813	45.7142	25.5814	0.0743
	BDLC	64.2857	65	41.9354	50.9803	0.2637	35.7142	34.2105	0.2489
	SDC	75.7142	85	54.8387	66.6667	0.5183	24.2857	50	0.4892
	SVM (RBF)	85.7142	85	70.8333	77.2727	0.6757	14.2857	62.9629	0.6698
Chi^2^pdf	NLR	61.4285	65	39.3939	49.0566	0.2262	38.5714	32.5	0.2092
	Linear Reg	72.8571	75	51.7241	61.2244	0.4310	27.1428	44.1176	0.4140
	Logistic Regression	70	80	48.4848	60.3773	0.4162	30	43.2432	0.3849
	GMM	64.2857	60	41.3793	48.9795	0.2384	35.7142	32.4324	0.2290
	BDLC	67.1428	65	44.8275	53.0612	0.3026	32.8571	36.1111	0.2907
	SDC	71.4285	70	50	58.3333	0.3872	28.5714	41.1764	0.375
	SVM (RBF)	84.2857	85	68	75.5556	0.6505	15.7142	60.7142	0.6418
Firefly Algorithm	NLR	55.7142	55	33.3333	41.5094	0.0995	44.2857	26.1904	0.0920
	Linear Reg	55.7142	55	33.3333	41.5094	0.0995	44.2857	26.1904	0.0920
	Logistic Regression	71.4285	85	50	62.9629	0.4609	28.5714	45.9459	0.4214
	GMM	58.5714	60	36.3636	45.2830	0.1628	41.4285	29.2682	0.1506
	BDLC	75.7142	70	56	62.2222	0.4525	24.2857	45.1612	0.4465
	SDC	54.2857	50	31.25	38.4615	0.0544	45.7142	23.8095	0.0508
	SVM (RBF)	85.7142	80	72.7272	76.1904	0.6617	14.2857	61.5384	0.6601
Cuckoo Search	NLR	62.8571	65	40.625	50	0.2448	37.1428	33.3333	0.2288
	Linear Reg	71.4285	70	50	58.3333	0.3872	28.5714	41.1764	0.375
	Logistic Regression	71.4285	70	50	58.3333	0.3872	28.5714	41.1764	0.375
	GMM	80	80	61.5384	69.5652	0.5609	20	53.3333	0.5504
	BDLC	58.5714	60	36.3636	45.2830	0.1628	41.4285	29.2682	0.1506
	SDC	68.5714	70	46.6666	56	0.3468	31.4285	38.8889	0.3304
	SVM (RBF)	**90**	**90**	**78.2608**	**83.7209**	**0.7694**	**10**	**72**	**0.7655**

**Table 12 diagnostics-13-02654-t012:** Computational Complexity of the Classifiers for Different Dimensionality Reduction Method without Feature Selection Methods.

Classifiers	With DFA DR Method	With Chi^2^ pdf DR Method	With Firefly Algorithm DR Method	With Cuckoo Search DR Method
NLR	O(2n^2^log2n)	O(2n^2^log2n)	O(2n^4^log2n)	O(2n^4^log2n)
Linear Reg	O(2nlog2n)	O(2n2log2n)	O(2n^3^log2n)	O(2n^3^log2n)
Logistic Regression	O(2n^2^log2n)	O(2n^2^log2n)	O(2n^3^log2n)	O(2n^3^log2n)
GMM	O(2n^3^log2n)	O(2n^3^log2n)	O(2n^4^log2n)	O(2n^4^log2n)
BDLC	O(2n3 log2n)	O(2n3log2n)	O(2n4log2n)	O(2n4 log2n)
SDC	O(2n^2^log2n)	O(2n^2^log2n)	O(2n^4^log2n)	O(2n^4^log2n)
SVM (RBF)	O(2n^2^log4n)	O(2n^2^log4n)	O(2n^4^log4n)	O(2n^3^log4n)

**Table 13 diagnostics-13-02654-t013:** Computational Complexity of the Classifiers for Different Dimensionality Reduction Method with PSO Feature Selection Method.

Classifiers	With DFA DR Method	With Chi^2^ pdf DR Method	With Firefly Algorithm DR Method	With Cuckoo Search DR Method
NLR	O(2n^4^log2n)	O(2n^4^log2n)	O(2n^6^log2n)	O(2n^6^log2n)
Linear Reg	O(2n^3^log2n)	O(2n^4^log2n)	O(2n^5^log2n)	O(2n^5^log2n)
Logistic Regression	O(2n^4^log2n)	O(2n^4^log2n)	**O(2n^4^log2n)**	O(2n^5^log2n)
GMM	O(2n^5^log2n)	O(2n^5^log2n)	O(2n^6^log2n)	O(2n^6^log2n)
BDLC	O(2n^5^log2n)	O(2n^5^log2n)	O(2n^6^log2n)	O(2n^6^log2n)
SDC	O(2n^4^log2n)	O(2n^4^log2n)	O(2n^4^log2n)	O(2n^4^log2n)
SVM (RBF)	O(2n^4^log2n)	**O(2n^4^log4n)**	O(2n^6^log4n)	O(2n^5^log4n)

**Table 14 diagnostics-13-02654-t014:** Computational Complexity of the Classifiers for Different Dimensionality Reduction Method with Harmonic Search Feature Selection Method.

Classifiers	With DFA DR Method	With Chi^2^ pdf DR Method	With Firefly Algorithm DR Method	With Cuckoo Search DR Method
NLR	O(2n^3^log2n)	O (2n^3^log2n)	O(2n^4^log2n)	O(2n^4^log2n)
Linear Reg	**O(2n^2^log2n)**	O(2n^3^log2n)	O(2n^5^log2n)	O(2n^5^log2n)
Logistic Regression	O(2n^3^log2n)	O(2n^3^log2n)	O(2n^4^log2n)	O(2n^4^log2n)
GMM	O(2n^4^log2n)	O(2n^5^log2n)	O(2n^5^log2n)	O(2n^5^log2n)
BDLC	O(2n^4^log2n)	O(2n^4^log2n)	O(2n^5^log2n)	O(2n^5^log2n)
SDC	O(2n^3^log2n)	O(2n^3^log2n)	O(2n^3^log2n)	O(2n^4^log2n)
SVM (RBF)	O(2n^4^log2n)	O(2n^4^log2n)	O(2n^4^log2n)	**O (2n^4^log2n)**

**Table 15 diagnostics-13-02654-t015:** Comparison with Previous Works.

S. No	Author (With Year)	Description of the Population	Data Sampling	Machine Learning Parameter	Accuracy (%)
1	Kumar et al. (2017) [52]	Diagnosis lab, Warangal—IN. Diabetes: 650	*N*-fold (*N* = 10) cross validation	Support Vector Machine, Naive Bayes, K-nearest neighbor C4.5 Decision tree	Accuracy: 69, 67, 70, 74
2	Olivera et al. (2017) [53]	Diabetes: 12,447 unknown: 1359 age: 35–74	Training set (70%) test set (30%) tenfold cross-validation	Logistic Regression Artificial Neural Network K-nearest neighbor Naïve Bayes	Balanced Accuracy: 69.3, 69.47, 68.74, 68.95
3	Alghamdi et al. (2017) [54]	Total: 32,555 diabetes: 5099 imbalanced	N-fold cross validation	Naïve Bayes tree, random forest, and logistic model tree, j48 decision tree	Accuracy: 83.9, 84.1, 79.9, 84.3, 84.1
4	Xie et al. (2017) [55]	Total: 21,285 diabetes: 1124 age: 35–65	Training set (75%) test set (25%)	K2 structure-learning algorithm	Accuracy = 82.48
5	Sarwar et al. (2018) [56]	Pima Indians Diabetes Dataset = 768	Training set (70%) test set (30%) tenfold cross-validation	K nearest neighbors, Naïve Bayes, support vector machine, decision tree, Logistic Regression, random forest	Accuracy: 77, 74, 77, 71, 74, 71
6	Zou et al. (2018) [57]	Physical Examination data in Luzhou, China. Total data: 164431	Fivefold cross-validation	Random forest J48 decision tree Deep Neural Network	Accuracy: 81, 79, 78
7	Perveen et al. (2019) [58]	Total: 667, 907 age: 22–74 diabetes: 8.13% imbalance	K-medoids under sampling	J48 decision tree, Naïve Bayes	Accuracy: 88.3, 87.3, 83.6, 82.6
8	Yuvaraj et al. (2019) [59]	Total: 75,664	Training set (70%) test set (30%)	Decision tree Naïve Bayes random forest	Accuracy: 88
9	Jakka et al. (2019) [60]	Pima Indians Diabetes dataset	None	K nearest neighbor, decision tree, Naive Bayes, support vector machine, Logistic Regression, random forest	Accuracy: 73, 70, 75, 66, 78, 74
10	Radja et al. (2019) [61]	Total: 768 diabetes: 500 control: 268	Tenfold cross-validation	Naive Bayes, support vector machine, decision table, J48 decision tree	Accuracy: 80, 79, 76, 79
11	Xiong et al. (2019) [62]	Total: 11845 diabetes: 845 age: 20–100	Training set (60%) test set (20%) tenfold cross-validation set (20%)	Multilayer perceptron, Ada-Boost, Random Forest, Support Vector Machine, Gradient boosting	Accuracy: 87, 86, 86, 86, 86
12	Dinh et al. (2019) [63]	Case 1: 21,131 diabetes: 5532 case 2: 16,426 prediabetes: 6482	Training set (80%) test set (20%) tenfold cross-validation	Support vector machine, random forest, gradient boosting, Logistic Regression	Accuracy: 89, 84, 86, 72
13	Yang et al. (2020) [64]	Total =8057 age: 20–89 imbalanced Non-Diabetic = 6721 Diabetic = 1336	Training set: (80%, 2011–2014), test set: (20%, 2011–2014) and validation set: (2015–2016) fivefold cross-validation	Linear discriminant analysis, support vector machine random forest	Accuracy: 75, 74, 74
14	Muhammad et al. (2020) [65]	Total: 383 age: 1–150 diabetes: 51.9%	Tenfold cross-validation	Logistic Regression Support vector machine K-nearest neighbor Random forest Naive Bayes Gradient boosting	Accuracy: 81, 85, 82, 89, 77, 86
15	Lam et al. (2021) [66]	Control: 19,852 diabetes: 3103 age: 40–69	Tenfold cross-validation	Random forest Logistic Regression extreme gradient boosting GBT	Accuracy: 86
16	De Silva et al. (2021) [67]	Total: 16,429 diabetes: 5.6% age: >20	Training set (30%) validation (30%) test set (40%)	Logistic Regression	Accuracy: 62
17	Kim et al. (2021) [68]	Total: 3889 diabetes: 746 age: 40–69	Fivefold cross-validation	Deep neural network, Logistic Regression, decision tree	Accuracy: 80, 80, 71
18	Ramesh et al. (2021) [69]	Pima Indians	Tenfold cross-validation	Support vector machine	Accuracy: 83
19	This article	Nordic islet transplantation programme	10-fold cross-validation	LR, NLR, LoR, GMM, BLDC, SDC, SVM (RBF)	Accuracy: 91

**Table 16 diagnostics-13-02654-t016:** Comparison of classifier performance with different datasets.

S. No.	Authors	Data Set Used	Machine Learning Models/Classifiers	Accuracy (%)
1	Methaporn et al. (2023) [70]	Department of Medical Services, Bangkok, Thailand, from 2019–2021	Support Vector Machine (SVM)	81.1
			Decision Tree (DT)	81.3
			k-Nearest neighbour (kNN)	81.7
			Random forest	88.2
2	Pannapa C et al. (2021) [71]	National Institute of Diabetes and Digestive and Kidney Diseases—PIMA	Random Forest	76.30
			SVM (L)	77.08
			SVM (P)	74.74
			SVM (RBF)	75.78
			kNN	74.87
		Thyroid—University of California Irvine (UCI)	Random Forest	96.74
			SVM (L)	96.27
			SVM (P)	91.16
			SVM (RBF)	95.81
			kNN	96.28
3	Kumari D et al. (2021) [72]	PIMA Indian Data set—May 2008	Logistic Regression	74.89
			Support vector	74.08
			k-Nearest	71.92
			Navie Bayes	74.12
			Decision tree	71.42
			Random forest	77.48
			Ada boost	75.32
			XG Boost	75.75
			Gradient Boost	75.32
			CatBoost	75.32
4	Hui Yang et al. (2021) [73]	Physical examination data from the EMR of Luzhou Municipal Health Commission in China from 2011 to 2017	Linear regression	80
			XG Boost	72
			Random Forest	80
5	Amin Ul Haq et al. (2020) [74]	Dataset of diabetes, taken from the hospital Frankfurt, Germany	ID3	99
			Ada Boost	98.5
			Random Forest	98.3

## Data Availability

The data that support the findings of this study are available from the corresponding author upon reasonable request.

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
