# Peer review of "Detection of Diabetes through Microarray Genes with Enhancement of Classifiers Performance"

_diagnostics, 2023, doi:10.3390/diagnostics13162654_

Round 1

Reviewer 1 Report (Previous Reviewer 2)

Noiw, authors modified their manuscript according to the reviewers' comments adequately.

Author Response

Dear Sir/Madam,

Thank you for your kind review. Please find the answers in the attachment. 

With regards,

Dinesh Chellappan

Reviewer 2 Report (New Reviewer)

The following points are essential for improving the quality and transparency of the paper.  • In terms of grammar, the English language of the paper should be improved. • The "Introduction" part of the study should be expanded, considering the research objectives, problems, and hypotheses. • The design of the study should be specified in the Materials and Methods section.  • Research guideline(s)/standard(s) appropriate to the study design should be reported in the paper text. • The primary output/endpoint variable(s)/measurement(s) of the study should be defined. • Statistical tests for hypothesis testing and their assumptions should be specified in the study's statistical analysis in the Materials and Methods section.  • Are the data subjected to pre-processing?  • How were extreme/outlier values in the data determined and resolved?  • What approaches were used to test the validity of the models? • Which metrics were used in the performance evaluation of the estimates of models/algorithms?  • How were the predictive models selected in this study? • Which method(s) was/were used to optimize the hyperparameters of models/algorithms? • How was the most suitable cut-off point determined using the receiver operator characteristic (ROC) curve analysis? • The number of current references on the subject of the study should be increased.  • The discussion section of the research can be expanded by supporting current studies to address the findings of other studies reported with the present findings.   The following points are essential for improving the quality and transparency of the paper.  • In terms of grammar, the English language of the paper should be improved. • The "Introduction" part of the study should be expanded, considering the research objectives, problems, and hypotheses. • The design of the study should be specified in the Materials and Methods section.  • Research guideline(s)/standard(s) appropriate to the study design should be reported in the paper text. • The primary output/endpoint variable(s)/measurement(s) of the study should be defined. • Statistical tests for hypothesis testing and their assumptions should be specified in the study's statistical analysis in the Materials and Methods section.  • Are the data subjected to pre-processing?  • How were extreme/outlier values in the data determined and resolved?  • What approaches were used to test the validity of the models? • Which metrics were used in the performance evaluation of the estimates of models/algorithms?  • How were the predictive models selected in this study? • Which method(s) was/were used to optimize the hyperparameters of models/algorithms? • How was the most suitable cut-off point determined using the receiver operator characteristic (ROC) curve analysis? • The number of current references on the subject of the study should be increased.  • The discussion section of the research can be expanded by supporting current studies to address the findings of other studies reported with the present findings.  

Author Response

Dear Sir/Madam,

Thank you for your review and encouraging comments towards the improvement of the article quality. Please find herewith the attachment of the answers. 

With Regards,

Dinesh Chellappan

Harikumar Rajaguru

Round 2

Reviewer 2 Report (New Reviewer)

 Accept in present form

This manuscript is a resubmission of an earlier submission. The following is a list of the peer review reports and author responses from that submission.

Round 1

Reviewer 1 Report

This manuscript addresses an important topic, which is considered to some as a global chronic disease pandemic: diabetes.

1. The writing needs to be streamlined and better focused on the specific research question at hand. The abstract and introduction is too broad and makes it very difficult to glean where there research gap is and how the present study will address that gap. 

2. The manuscript reads like a thesis or grant proposal rather than a manuscript. I would suggest breaking it up into two or three papers, each with a specific focus.

3. I did not have a clear idea of what the research question that was driving the analytical approach.

4. I am confused by section 2.2 Organization of the paper. Does not seem to fit what is presented.

5. I think the authors could have provided a better description of the sample and context of the current study. They discuss in the introduction India and higher risk of diabetes but not really sure who the sample was, from where, and what are some characteristics that are important for the reader to know. 

6. The authors suggest that this analysis will help inform early detection of diabetes or pre-disease signs that could be detected and acted upon to prevent onset. Since the samples came from cadaver pancreas, I am wondering how this analysis would translate into non-invasive real world detection.

7. There are a substantial number of table and figures describing analysis without any real rationale of why they are included and how do the various analyses contribute to the underlying research question.

There are structural issues with sentences throughout sometimes making it difficult to understand the idea being expressed. 

Reviewer 2 Report

This paper presents the authors' attempt to improve the accuracy of classification as diabetic or non-diabetic by means of MicroArray genetic analysis in detecting diabetes.

The content of the paper is not inadequate.

However, even after reading the introduction and conclusion, it is not clear how this attempt should be applied to actual clinical practice or what concrete measures should be taken.

This is especially true if the source of the microarray is pancreatic tissue.

The addition of this point of view is required.